# GVCCS: A Dataset for Contrail Identification and Tracking on Visible Whole Sky Camera Sequences

Gabriel Jarry[1], Ramon Dalmau[1], Philippe Very[1], Franck Ballerini[1], and Stefania-Denisa Bocu[1]

[1]EUROCONTROL. Aviation Sustainability Unit (ASU), Aerodrome Centre Bois des Bordes, Brétigny-Sur-Orge, 91220, Essone, France

**Correspondence:** Gabriel Jarry (gabriel.jarry@eurocontrol.int)

**Abstract.** Aviation's climate impact includes not only $CO_2$ emissions but also significant non-$CO_2$ effects, especially from contrails. These ice clouds can alter Earth's radiative balance, potentially rivaling the warming effect of aviation $CO_2$. Physics-based models provide useful estimates of contrail formation and climate impact, but their accuracy depends heavily on the quality of atmospheric input data and on assumptions used to represent complex processes like ice particle formation and humidity-driven persistence. Observational data from remote sensors, such as satellites and ground cameras, could be used to validate and calibrate these models. However, existing datasets do not explore all aspects of contrail dynamics and formation: they typically lack temporal tracking, and do not attribute contrails to their source flights. To address these limitations, we present the Ground Visible Camera Contrail Sequences (GVCCS), a new open data set of contrails recorded with a ground-based all-sky camera in the visible range. Each contrail is individually labeled and tracked over time, allowing a detailed analysis of its lifecycle. The dataset contains 122 video sequences (24,228 frames) and includes flight identifiers for contrails that form above the camera. As reference, we also propose a unified deep learning framework for contrail analysis using a panoptic segmentation model that performs semantic segmentation (contrail pixel identification), instance segmentation (individual contrail separation), and temporal tracking in a single architecture. By providing high-quality, temporally resolved annotations and a benchmark for model evaluation, our work supports improved contrail monitoring and will facilitate better calibration of physical models. This sets the groundwork for more accurate climate impact understanding and assessments.

## 1 Introduction

Aviation contributes to global climate change not only through carbon dioxide ($CO_2$) emissions but also through a variety of non-$CO_2$ effects, including nitrogen oxides ($NO_x$), water vapour, and aerosols. Among these, condensation trails (contrails), ice-crystal clouds formed by aircraft at typical cruising altitudes, stand out for their potentially large yet uncertain radiative impact. Though they often appear as ephemeral white streaks in the sky, persistent contrails can spread into extensive cirrus-like cloud formations that reduce outgoing long-wave radiation, warming the planet. Recent studies suggest that the climate forcing from contrail cirrus is of the same order of magnitude as that from aviation $CO_2$ emissions (Lee et al., 2021; Teoh et al., 2023), although this comparison depends on the metric used (Borella et al., 2024).

Accurately assessing the climate impact of contrails remains a significant challenge for both aviation and climate scientists. Contrail lifecycles depend on complex interrelated processes, including ice nucleation, crystal growth, wind-driven dispersion, and interaction with natural clouds, that are sensitive to ambient atmospheric conditions. Small variations in temperature and humidity, particularly relative humidity with respect to ice, can determine whether a contrail dissipates quickly or persists and spreads. This sensitivity, combined with diurnal variability in radiative forcing (daytime cooling from reflected sunlight versus nighttime warming from trapped infrared radiation), makes the net climate effect of contrails highly variable and challenging to model.

While contrail impacts have traditionally been studied using physical models, recent advances in remote sensing and computer vision now offer a valuable observational perspective. Physics-based models, such as the Contrail Cirrus Prediction model (CoCiP) (Schumann, 2012) and the Aircraft Plume Chemistry, Emissions, and Microphysics Model (APCEMM) (Fritz et al., 2020), simulate contrail lifecycles by solving complex equations that describe interactions between aircraft emissions and atmospheric conditions. These models provide valuable theoretical insights, but their accuracy depends heavily on the quality of input data (Gierens et al., 2020). Key parameters, such as atmospheric temperature, humidity, and aircraft engine characteristics, are often uncertain, and these uncertainties propagate through calculations, affecting result reliability. Moreover, detailed simulations of contrail microphysics and radiative effects can be computationally demanding, particularly when applied to global-scale analyses.

Observational methods using satellite and ground-based imagery offer a direct, data-driven approach to studying contrails that complements theoretical models. Satellite-based contrail detection has a long history, beginning with early automated methods that leveraged brightness temperature differences and Hough transforms in NOAA-AVHRR imagery (Mannstein et al., 1999). Subsequent work extended these techniques to study regional radiative forcing (Meyer et al., 2002), contrail coverage and properties (Minnis et al., 2005; Palikonda et al., 2005; Mannstein and Schumann, 2005), and global contrail distributions (Meyer et al., 2007). Advances in sensor technology, particularly with MSG/SEVIRI, enabled rapid-scan observations that facilitated automated contrail tracking (Vázquez-Navarro et al., 2010), lifecycle analysis (Vázquez-Navarro et al., 2015), and improved detection algorithms (Ewald et al., 2013; Mannstein et al., 2012). Ground-based validation campaigns (Mannstein et al., 2010; Schumann et al., 2013) provided essential verification of satellite-derived contrail properties. More recently, high-resolution remote sensing combined with modern computer vision and deep learning has further enhanced detection capabilities (Meijer et al., 2022; McCloskey et al., 2021; Ng et al., 2023; Chevallier et al., 2023).

Despite growing interest in observational contrail analysis, publicly available datasets remain limited in scope. Existing datasets of contrails annotated in observational data, such as Google's OpenContrails, do not track individual contrails over time or provide information on the flights that formed them. Specifically, OpenContrails offers instance-level masks only on the central GOES-16 frame, with surrounding images left unannotated. In contrast, Sarna et al. (2025) introduced SynthOpenContrails, which overlays synthetic contrails and annotations onto real scenes, providing full per-frame localization, tracking, and flight attribution. This demonstrates that richly annotated data can exist, even if confined to synthetic contrail overlays rather than human annotation. An ideal scenario would be a fully annotated video dataset where every frame is labeled and each contrail is assigned a persistent identifier across time.

To advance research in this area, we present the Ground Visible Camera Contrail Sequences (GVCCS), an open dataset (Jarry et al., 2025) with instance-level annotations, derived from ground-based video recordings in Brétigny-sur-Orge, France (Réuniwatt CamVision visible ground-based camera). Our dataset includes 122 videos (of duration between 20 minutes and 5 hours) with a total of approximately 24,200 frames, each annotated with instance-level labels. By making this dataset openly available, we provide a valuable benchmark for both the atmospheric and aviation research communities.

To support future performance comparisons, we introduce a deep learning-based model for contrail segmentation and tracking. Instead of relying on separate models for these tasks, an approach that often requires complex, ad-hoc combinations of techniques, we adopt a unified framework based on Mask2Former (Cheng et al., 2021b), a state-of-the-art computer vision model. Mask2Former is designed for panoptic segmentation, which combines semantic segmentation (labeling each pixel with a class, e.g., "contrail" or "sky") and instance segmentation (distinguishing between individual objects, e.g., different contrails). In addition to separating contrails from clear sky, it can handle complex backgrounds, such as low-altitude cloud layers that partially or fully obscure contrails, by assigning appropriate "cloud" labels while still maintaining unique instance identities. For example, in a single image, panoptic segmentation can identify all visible contrail pixels, correctly label intervening clouds, and assign consistent instance masks to each contrail, even when they overlap, intersect, appear fragmented, or are seen through thin cloud cover. In fact, contrails often break into multiple disconnected components due to atmospheric conditions and natural dissipation processes. A robust monitoring system must not only identify these fragments but also associate them with the correct contrail instance.

It is worth noting that fragmentation poses a significant challenge for contrail analysis based solely on images or videos: visually disjointed segments from the same flight must be grouped without external data. Moreover, low-altitude cloud obscuration and sun glare can further interrupt or mask contrail continuity, producing multi-polygon annotations even for a single physical contrail. In operational settings, however, it is possible to first perform single-polygon instance segmentation and then associate multiple instances with the same flight using auxiliary data such as aircraft trajectories and wind fields. This post-processing step enables grouping across time and space based on flight identity rather than visual continuity. In this work, we restrict ourselves to purely image-based analysis and defer the integration of external data sources to future work.

Mask2Former, originally designed for individual images, can be easily extended to video data to improve the consistency of panoptic segmentation across frames (Cheng et al., 2021a). By leveraging temporal information, Mask2Former for videos performs semantic segmentation, instance segmentation, and tracking in an integrated manner. In this paper, we study both the frame-based and video-based versions of Mask2Former, comparing their performance on our dataset.

The remainder of this paper is structured as follows. Section 2 provides the necessary background on contrail formation and computer vision techniques, establishing the foundation for the challenges addressed in this work. Section 3 reviews related work on contrail datasets and segmentation models, highlighting current limitations and motivating our approach. Section 4 introduces our newly developed video-based dataset, detailing its annotation methodology and unique instance-level structure. Section 5 describes our panoptic segmentation framework based on the Mask2Former architecture. Section 6 presents and analyses the experimental results. Finally, Section 7 summarises our main contributions and outlines future research directions.

## 2 Background

This section introduces the key concepts necessary to understand the challenges addressed in this work. We begin by outlining the physical processes behind contrail formation and their implications for climate, focusing on why contrails are particularly difficult to detect and track. We then review relevant computer vision techniques, specifically object detection and image segmentation, and assess their suitability for analysing contrails.

### 2.1 The Science of Contrails

Contrails are artificial clouds that form behind aircraft when hot, humid engine exhaust mixes with the cold, low-pressure air at cruising altitudes, typically between 8 and 12 km. If atmospheric conditions are suitable — specifically, if the temperature falls below a critical threshold (typically around $-40\,°C$, depending on pressure and humidity) and the air is sufficiently humid — the water vapour in the exhaust condenses and freezes into ice crystals. The physical mechanism underlying this process was first explained by Schmidt (1941), who recognized that contrails form when ambient temperature is low enough to cause the humidity inside the aircraft plume to reach saturation with respect to liquid water, triggering condensation. Appleman (1953) provided further quantitative analysis, though without fully accounting for engine characteristics. Schumann (1996) later developed a comprehensive treatment incorporating engine efficiency and practical application methods, formalizing what is now known as the Schmidt–Appleman criterion. This process produces the familiar thin, white trails visible in the sky.

Like natural clouds, contrails influence Earth's radiation budget: they reduce outgoing long-wave radiation, leading to warming, while also reflecting incoming solar radiation, which has a cooling effect. The net result depends on the contrail's altitude, optical properties, lifespan, and time of day. The magnitude of contrail climate forcing relative to aviation's $CO_2$ emissions depends on the climate metric chosen (Borella et al., 2024); however, contrails are thought to warm the climate at a level of the same order of magnitude as aviation's $CO_2$ emissions (Lee et al., 2021; Teoh et al., 2023). This makes the monitoring and characterization of contrails an essential part of understanding aviation's full environmental footprint (Teoh et al., 2023) and developing mitigation strategies (Teoh et al., 2020).

Quantifying this radiative forcing requires understanding both contrail optical properties and their spatial and temporal distribution. Early satellite-based studies provided first estimates of regional contrail radiative effects (Meyer et al., 2002) and developed parametric models linking contrail properties to radiative forcing (Schumann et al., 2009). Climatological analyses of persistent contrails revealed dependencies on atmospheric conditions and aircraft traffic patterns (Iwabuchi et al., 2012; Mannstein and Schumann, 2005), while ground-based observations offered validation of satellite-derived contrail properties (Mannstein et al., 2010).

As mentioned above, the observational viewpoint offers an alternative perspective that focuses on detecting and analysing contrails directly using satellite and ground-based remote sensing instruments. However, detecting and tracking contrails presents several technical challenges, which helps explain the growing research interest in the topic. Satellite imagery often lacks the spatial and temporal resolution needed to detect contrails in their early stages (Ng et al., 2023; Mannstein et al., 2010). Geostationary satellites have a nominal spatial resolution of about 0.5 to 2 km and a temporal resolution of 5 to 15 min,

which is often insufficient to capture the narrow, faint, and short-lived nature of freshly formed contrails unless they persist and grow. Even when contrails spread into detectable cloud structures, they are difficult to distinguish from natural cirrus, particularly in scenes with complex cloud layers. Moreover, by the time a contrail is visible in satellite images, it has often drifted and deformed, complicating attribution to the flight that produced it (Chevallier et al., 2023; Sarna et al., 2025). This linkage is crucial, as identifying the originating flight enables researchers to retrieve essential details such as aircraft type and engine model, key inputs for assessing contrails' environmental impact and improving physical models through comparison with empirical observations.

Ground-based cameras (Schumann et al., 2013; Low et al., 2025) offer a complementary perspective with critical advantages. Positioned beneath flight paths, these systems can capture high-resolution images and video with far greater spatial and temporal fidelity than satellites. Crucially, they can detect contrails immediately after formation, while they are still thin, linear, and visually distinct. This early visibility simplifies the task of associating observed contrails with the specific flight responsible, especially when combined with precise trajectory data. The main drawback is, naturally, their restricted spatial coverage, which hinders the ability to monitor contrails from formation to dissipation.

This attribution advantage is particularly significant compared to satellite-based approaches. Geostationary satellites face several challenges: their coarse spatial resolution ($\sim$0.5–2 km/pixel) means contrails must persist and spread before becoming detectable, by which time wind advection has displaced them substantially from their formation location; their temporal resolution (5–15 minutes) means the originating aircraft may be far away when the contrail first appears; and multiple aircraft may have traversed similar airspace during this window, creating ambiguity. Attribution from satellite data therefore requires sophisticated algorithms accounting for wind fields, parallax, and probabilistic matching (Chevallier et al., 2023; Riggi-Carrolo et al., 2023; Geraedts et al., 2024; Sarna et al., 2025). In contrast, ground-based cameras observe contrails at formation with high spatial resolution ($\sim$73 m/pixel at 10 km altitude in our system) and 30-second sampling, enabling straightforward contrail-to-flight attribution without the ambiguities inherent in satellite-based approaches.

While not the focus of this paper, one promising direction involves combining ground-based and satellite observations into a unified monitoring framework. In such a system, contrails would first be detected in high-resolution ground-based imagery and attributed to specific flights using trajectory and weather data, providing access to key aircraft and engine parameters. Crucially, to enable continuous tracking beyond the limited field of view of the ground-based camera, these contrails would then need to be reliably linked to their evolving counterparts in satellite imagery as they drift, expand, and age. Successfully associating contrails across these two modalities — ground and satellite — would allow monitoring of their full lifecycle from formation to dissipation while preserving information about the specific aircraft and flight responsible for creating them.

## 2.2 Computer Vision Techniques for Contrail Monitoring

Contrails are visually challenging targets for computer vision due to their thin, elongated shapes, variable curvature, and tendency to fragment or fade over time. These characteristics make them fundamentally different from the objects typically addressed in standard object detection benchmarks, such as vehicles and animals in datasets like the Common Objects in Context (COCO) dataset (Lin et al., 2014), which features well-defined, discrete objects.

Traditionally, object detection methods localise targets using bounding boxes, usually axis-aligned rectangles. Standard approaches such as Faster R-CNN (Ren et al., 2017) and YOLO (Redmon et al., 2016) exemplify this paradigm. This approach works well for objects like cars or animals, which are compact and roughly rectangular, but performs poorly for contrails. A single axis-aligned bounding box may inadvertently include multiple contrail segments or large amounts of background sky, while missing parts of curved or fragmented trails. Oriented bounding boxes offer some improvement by allowing rotation,

which better fits the geometry of elongated contrails. However, they still fall short in capturing fine-grained shapes, gaps, or fading segments. Figure 1 shows the limitations of axis-aligned and oriented bounding boxes for object detection on contrails.

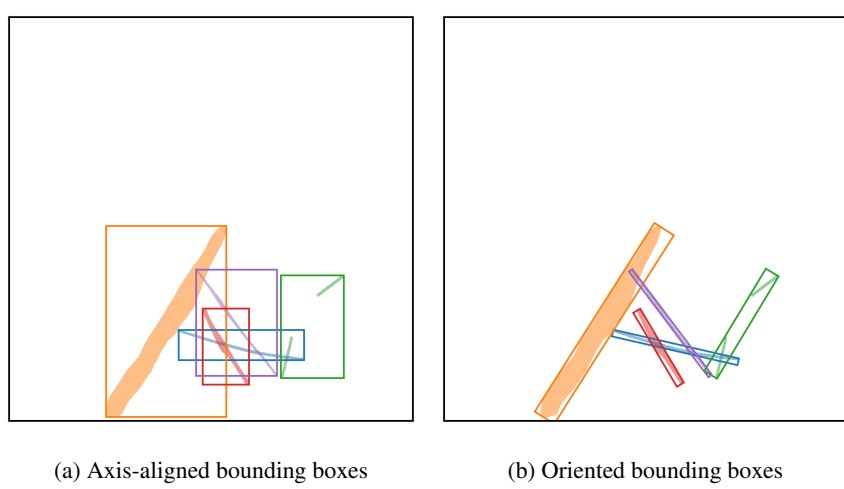

(a) Axis-aligned bounding boxes         (b) Oriented bounding boxes

**Figure 1.** Illustration of bounding box detection on contrails. Each detected contrail is highlighted with a distinct color. Note how elongated or fragmented trails challenge bounding box alignment and separation.

Instance segmentation provides a more precise solution by predicting pixel-level masks for each individual object. This approach is particularly beneficial for contrails, as it can delineate each trail accurately even when they intersect, overlap, or dissipate unevenly. For instance, two overlapping contrails that fade at different rates can still be assigned to distinct instances.

It is important to note that instance segmentation has been addressed in atmospheric science for decades using classical computer vision techniques. Early work by Mannstein et al. (1999) detected contrail pixels and grouped spatially connected regions into distinct objects. Similarly, Schumann et al. (2013) used ground-based cameras with automated algorithms to identify, track, and characterize individual contrails. These methods achieved instance-level contrail separation through feature-based detection, connectivity analysis, and trajectory matching. Our work builds on this foundation by applying modern deep

learning architectures that perform instance segmentation through learned feature representations rather than hand-crafted rules.

Semantic segmentation, in contrast, labels each pixel by class (e.g., "contrail" or "sky") but does not distinguish between individual contrails. This is insufficient when studying temporal evolution or interactions between specific contrails, since it treats all contrails as a single undifferentiated class.

Panoptic segmentation combines the strengths of both approaches: it assigns a class label to every pixel (semantic segmentation) and an instance identifier where appropriate (instance segmentation). In this framework, "things" such as individual contrails are assigned unique instance labels, while "stuff" like the background sky or natural clouds is labelled only by class. This unified view is well-suited to contrail monitoring, enabling fine-grained analysis of individual contrails within the broader atmospheric context. Moreover, the framework can be readily extended to additional classes (e.g., cirrus, cumulus) for more comprehensive scene understanding, provided these classes have been effectively and consistently labelled during dataset creation, which introduces an additional layer of complexity to the annotation campaign. Figure 2 illustrates the instance, semantic, and panoptic segmentation methods.

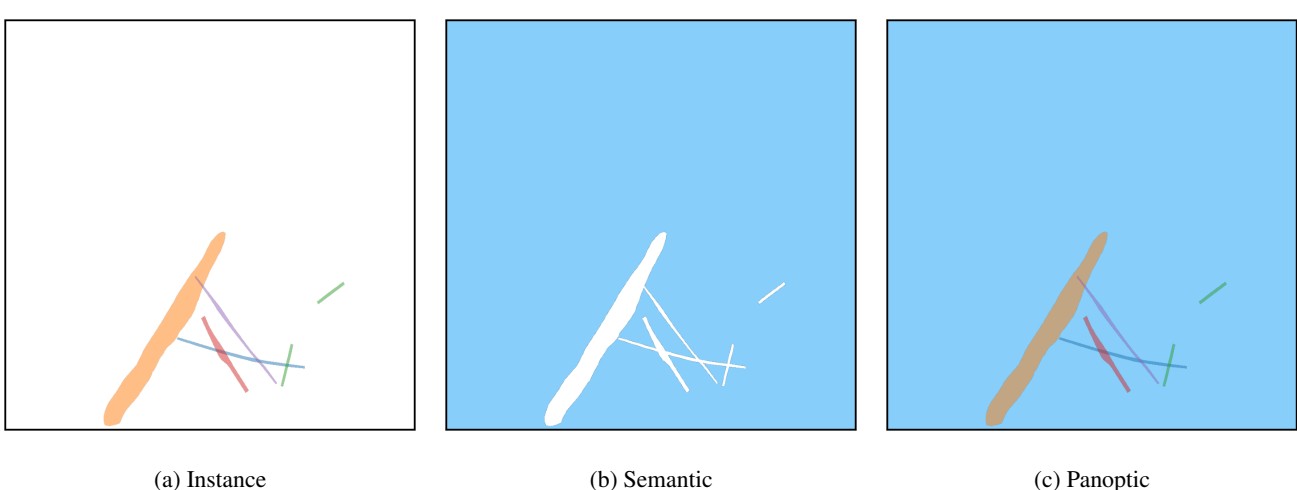

    (a) Instance             (b) Semantic             (c) Panoptic

**Figure 2.** Comparison of segmentation methods applied to illustrative contrails. (a) Instance segmentation assigns unique identifiers (colors) to each contrail, enabling individual tracking but without classifying non-contrail regions. (b) Semantic segmentation identifies all contrail pixels as a single class (white) versus background (blue), without distinguishing between individual contrails. (c) Panoptic segmentation combines both approaches: each contrail receives a unique identifier while all pixels are classified (contrails in color, background in blue). This unified representation enables both instance-level tracking and scene-level understanding.

## 3  State of the Art

This section presents an overview of prior work in contrail segmentation and analysis, focusing first on the datasets that have been developed to support this research, and then on the computational models used for contrail segmentation and flight attribution. The scope and key features of existing datasets are outlined, with particular attention given to the limited availability of temporal annotations and flight attribution ground truth. Subsequently, we examine state-of-the-art segmentation and tracking methods, particularly deep learning-based approaches, assessing their applicability and performance in contrail analysis. This review highlights gaps in current research and motivates the contributions presented in this paper.

## 3.1  Datasets

Recent advances in contrail detection have been supported by the development of annotated datasets, primarily based on satellite imagery. These datasets have facilitated the application of computer vision techniques for contrail identification, although aspects such as temporal continuity and integration with flight metadata remain limited in most cases. In this section, we review the most relevant publicly available datasets and place our contributions within this context.

Kulik (2019) and Meijer et al. (2022) are, to our knowledge, the first studies to leverage a modern, data-driven, deep learning framework for large-scale contrail segmentation. The authors developed and applied convolutional neural networks, which were trained using a manually curated dataset comprising over 100 annotated geostationary GOES satellite images with instance segmentation.

One of the first large-scale labelling efforts in contrail detection was led by Google Research, beginning with the development of a contrail dataset based on high-resolution Sentinel satellite imagery (McCloskey et al., 2021). Human experts manually annotated the images using structured guidelines, producing masks that identify contrail pixels at the semantic segmentation level, distinguishing contrail from non-contrail regions without tracking individual contrail instances. Multiple annotators independently labelled each image, and the dataset includes all individual annotations, with the option to filter results by majority consensus. This methodology improved both the spatial precision and overall quality of the labels.

Building on this work, Google released the OpenContrails dataset (Ng et al., 2023), which is based on images from the GOES-16 Advanced Baseline Imager (ABI). OpenContrails offers temporal context by including short sequences of unlabelled images surrounding each annotated frame, providing valuable information to annotators for more accurate labelling. Only the central frame in each sequence is annotated, therefore not allowing direct comparison of contrail dynamics with physical models.

In the domain of ground-based data for contrail research, significant resources have been developed to support computer vision tasks. Gourgue et al. (2025) introduce an open-access corpus of around 1,600 polygon-annotated hemispheric sky images acquired at the SIRTA atmospheric laboratory, near Paris, offering class labels that distinguish "young," "old," and "very old" contrails as well as several confounding artefacts. By capturing high-resolution ground views minutes after formation, this dataset fills a temporal–spatial gap left by satellite benchmarks. Complementary to this data provision, Pertino et al. (2024) focus on the development of detection methodology, providing a comprehensive comparison of computer vision models applied to both visible and infrared images.

Rather than creating a dataset for training modern convolutional networks on segmentation tasks, Low et al. (2025) manually annotated the correspondence between contrail waypoints derived from the application of the CoCiP model and observations from their wide-angle ground camera system. This approach is particularly well-suited for directly assessing and parametrizing physical models.

Earlier studies have successfully collocated contrails using various combinations of sensors, including ground-based observations, satellite imagery, and lidar data (Iwabuchi et al., 2012; Mannstein et al., 2010). For example, Vázquez-Navarro et al. (2010) demonstrated tracking contrails first identified in high-resolution MODIS imagery through time sequences of Meteosat

data, leveraging complementary spatial and temporal resolution. Building on this foundation, Meijer et al. (2024) is, to our knowledge, the first example of a dataset specifically designed for contrail altitude estimation by collocating images from two distinct remote sensors: they assembled a dataset comprising over 3,000 cases over the contiguous United States (2018–2022). Contrails were first located via automated detection in GOES-16 ABI infrared imagery, then precisely collocated, correcting for parallax and wind advection, with CALIOP lidar cross-sections. The team then conducted manual inspections of the matched imagery to verify and validate alignment. This benchmark dataset linking geostationary contrail signatures to high-resolution vertical profiles enables supervised deep-learning approaches to predict contrail top heights from ABI data.

A significant advance in contrail detection has been the development of synthetically labelled datasets. Chevallier et al. (2023) generated a synthetic dataset using CoCiP (Schumann, 2012) to overlay contrail polygons onto GOES-16 imagery, enabling the first instance segmentation pipeline for contrail detection. The performance of flight assignment algorithms was validated using actual GOES data through manual inspection rather than synthetic reference ground truth. Building on this synthetic foundation, Sarna et al. (2025) introduced a benchmark dataset, SynthOpenContrails, with sequences of synthetic contrail detections tied to known flight metadata, providing the first opportunity to quantitatively evaluate and improve con­trail–flight attribution algorithms. To our knowledge, this is the only dataset providing localized and tracked contrails with attributable ground truth, albeit synthetic. While the use of synthetic datasets represents a modern and cutting-edge technique for training algorithms, the use of manually labelled data as test sets is theoretically preferable to objectively assess algorithmic performance. However, obtaining such datasets on geostationary satellite images, with their coarse resolution, remains very difficult at this stage, which motivates the approach adopted by the authors. As mentioned in Sarna et al. (2025), obtaining such a reference dataset with ground truth for flight attribution based on human annotations is feasible in principle with higher resolution low-orbit satellites or ground-based cameras, which is the focus of the present work.

Overall, while existing datasets have contributed valuable resources, there is a lack of comprehensive, human-labelled data containing temporally resolved, instance-level, and flight-attributed annotations. Our work addresses this issue by introducing a dataset designed to provide these annotations, collected using our ground camera system.

## 3.2 Models

Contrail monitoring with computer vision was first pioneered in the early nineties (Forkert et al., 1993; Mannstein et al., 1999), using traditional image-analysis techniques. Their work applied linear-kernel methods, direct thresholding of bright­ness temperature difference channels, and early Hough-transform operators (Pratt, 2007) optimized for linear shape detection to identify contrails in AVHRR satellite imagery. This foundational work was extended through improved detection algo­rithms (Meyer et al., 2002, 2007), automated tracking methods (Vázquez-Navarro et al., 2010), and enhanced cirrus detection capabilities (Ewald et al., 2013; Mannstein et al., 2012). Parallel advances in cloud property retrieval from geostationary satel­lites (Bugliaro et al., 2012; Hamann et al., 2014; Kox et al., 2014) and neural network-based classification (Strandgren et al., 2017b, a) further refined contrail and cirrus characterization. Ground-based validation studies (Mannstein et al., 2010; Schu­mann et al., 2013) provided essential verification of these satellite-based methods. These classical computer vision approaches were later complemented by improvements from Duda et al. (2013) and eventually by modern deep learning techniques.

To the best of our knowledge, Kulik (2019) and Meijer et al. (2022) represent the earliest applications of modern convolutional networks to pixel-level classification and semantic segmentation. Building on the OpenContrails dataset, Ng et al. (2023) employed semantic segmentation algorithms, specifically DeepLabV3 (Chen et al., 2017, 2018), to identify contrails in ash-RGB composites using brightness temperature differences. Their work demonstrated that adding temporal context via a 3D encoder, incorporating the time dimension, led to improved performance. Moreover, results from the subsequent Kaggle competition showed that U-Net models (Ronneberger et al., 2015) equipped with modern transformer backbones, such as MaxViT (Tu et al., 2022) and CoatNet (Dai et al., 2021), achieved even stronger results (Jarry et al., 2024).

Using an ensemble approach, Ortiz et al. (2025) combined six neural networks, including U-Net, DeepLab, and transformer architectures, and applied optical-flow-based corrections to maintain temporal consistency across consecutive satellite frames. Meanwhile, Sun and Roosenbrand (2025) introduced a Hough-space line-aware loss for few-shot scenarios, supplementing Dice loss with a global alignment term to encourage predictions to align with linear structures.

Shifting from pixel-level masks to instance-level contrail segmentation and making use of synthetic data, Chevallier et al. (2023) introduced the first algorithmic pipeline focused on instance segmentation for contrail detection, utilizing the Mask R-CNN algorithm (He et al., 2017). Similarly, Van Huffel et al. (2025) adopted Mask R-CNN to process images captured by their wide-angle ground camera system.

The challenging task of attributing detected contrails to individual flights in geostationary satellite imagery, typically using automatic dependent surveillance-broadcast (ADS-B) data, has been the focus of several recent studies. Chevallier et al. (2023) introduced a pipeline that combines contrail detection, tracking, and matching with aircraft using geometric criteria and wind-corrected trajectories. Riggi-Carrolo et al. (2023) proposed a probabilistic matching method that accounts for uncertainties in flight data and atmospheric conditions, incorporating features derived from Hough-based line detection to improve alignment. Geraedts et al. (2024) presented a scalable system designed to assign contrails to flights on a large scale, enabling routine monitoring of contrail formation and supporting climate assessments. Sarna et al. (2025) systematically benchmarked and refined these attribution algorithms, highlighting common challenges and proposing improved association metrics, building on the release of the synthetically generated SynthOpenContrails dataset.

By contrast, our work targets ground-based imagery, capturing contrails immediately after formation and enabling near-instantaneous flight attribution via ADS-B data. We harness panoptic segmentation using Mask2Former, trained on high-resolution video, to extract pixel-accurate masks of individual contrails and track them over time. This fills the gap in early-stage contrail detection and provides richer spatial and temporal detail than existing satellite-based models.

## 4 Dataset

The primary contribution of this paper is the introduction of a new dataset designed to support contrail detection, tracking, and attribution. This section provides a detailed overview of the dataset. Section 4.1 describes the data collection and labelling campaign. Section 4.2 summarizes the structure and content of the dataset.

## 4.1 Data collection and labelling campaign

To support the development of machine learning models for contrail detection, we conducted an extensive labelling campaign as part of the *ContrailNet* project. Visible-spectrum image sequences were acquired using an all-sky ground-based camera installed on the roof of the EUROCONTROL Innovation Hub (Location: $48°36'1.87''$ N, $2°20'48.46''$ E). The camera captured the sky every 30 seconds at a resolution of $1976 \times 2032$ pixels.

Our camera provider, Reuniwatt, delivered a dual all-sky camera system: the first unit, CamVision, operates in the visible spectrum, capturing high-resolution fisheye images every 30 seconds with on-board processing and self-calibration, ensuring reliable daytime operation even in dusty or wet conditions. The second unit, SkyInsight, uses long-wave infrared (8–13 $\mu$m) imaging via a chrome-coated hemispherical mirror and will be used in future research.

The raw all-sky images were first geometrically projected onto a square grid. This projection process uses camera-specific calibration files to associate each pixel with its corresponding azimuth and zenith angles, effectively removing lens distortions and re-mapping the sky onto a uniform Cartesian representation. A 75 km $\times$ 75 km grid of georeferenced points was computed at a fixed cloud altitude (10 km), and a linear interpolation scheme was used to assign raw pixel values to the projected frame. The output is a square image of size $1024 \times 1024$ pixels that preserves the spatial geometry of the sky above the camera.

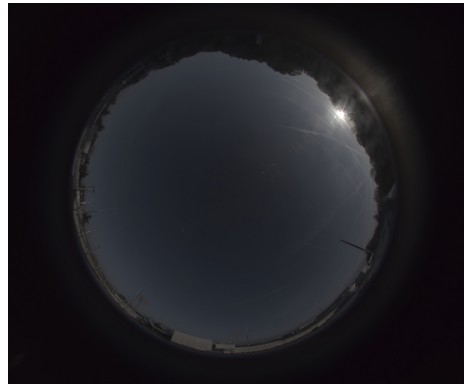 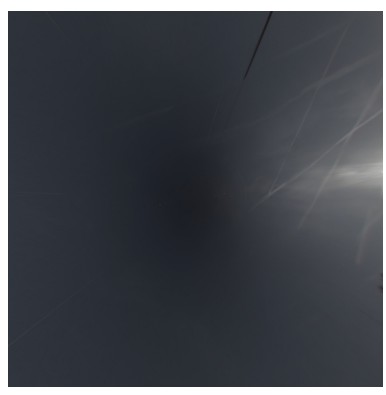 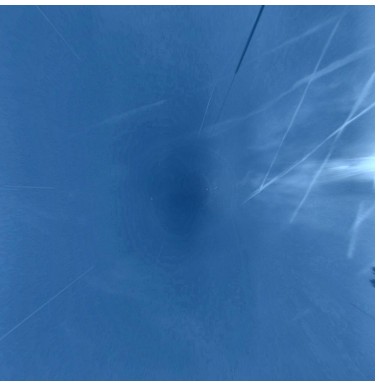

(a) Raw all-sky image showing severe fisheye distortion

(b) Geometric projection onto a square grid

(c) Final three-step enhancement process

**Figure 3.** Impact of preprocessing pipeline on contrail visibility.

To improve the visual clarity and consistency of the sequences, each projected image undergoes a three-step enhancement process. First, brightness is increased using a linear scaling operation to compensate for underexposure in certain atmospheric conditions. Second, local contrast is enhanced via CLAHE (Contrast Limited Adaptive Histogram Equalization), which boosts fine features like faint or fragmented contrails without overexposing bright regions. Finally, colour warmth is reduced by rebalancing the blue and red channels, mitigating the effects of high solar glare and improving contrail visibility in challenging lighting conditions. This preprocessing pipeline proved essential for two reasons: it enables annotators to identify and delineate contrails consistently across diverse atmospheric scenes, and it simplifies the learning task for computer vision models by

removing camera-specific distortions and enhancing the natural linear structure of contrails. Figure 3 illustrates the impact of geometric projection and enhancement, demonstrating how the pipeline reveals contrails that would otherwise be difficult or impossible to annotate reliably. All models presented in this work are trained and evaluated exclusively on preprocessed images.

The video sequences included in the dataset were not randomly sampled from the full archive. To ensure sufficient contrail instances for effective model training while maintaining seasonal and atmospheric diversity, we applied a two-stage selection strategy. First, the complete year-long archive was processed using a lightweight binary classifier to distinguish contrail-present from contrail-absent images. This automated filtering efficiently identified candidate periods by excluding extended intervals of clear sky or heavy low-altitude cloud cover. Second, video sequences were manually selected from these filtered periods, prioritizing scenes with visible, persistent contrails suitable for detailed temporal annotation. This approach deliberately over-samples contrail-positive cases, enhancing the dataset's utility for segmentation and tracking but introducing a selection bias that should be considered when evaluating model performance on unfiltered operational data. The final dataset spans the full calendar year, ensuring coverage of diverse seasonal and atmospheric conditions.

The labelling process was applied to video sequences; each sequence comprised between 60 and 480 images, corresponding to durations of 30 minutes to 4 hours, enabling the temporal tracking of contrails throughout their formation and dissipation phases.

The labelling process was carried out using a dedicated annotation tool developed by Encord, who also provided a professional team of annotators. We maintained close collaboration with this team through regular coordination meetings, during which the annotation guide was developed and iteratively refined. The labelling platform was specifically configured to overlay flight trajectory data above the camera's field of view, assisting annotators in identifying "new" contrails—those forming above the camera and visibly associated with a known aircraft trajectory. In contrast, "old" contrails were defined as those already present at the start of a sequence or likely formed outside the camera's field of view, making flight association impossible.

Each contrail was annotated using high-precision polygons that tracked its spatial extent throughout its visible evolution, from early linear stages to advanced spreading phases. When contrails became fragmented or partially obstructed by clouds, multiple polygons were used and linked using relational attributes ("fragmented contrail" and "cloud obstruction") to preserve temporal continuity.

To ensure the highest annotation quality, the campaign incorporated a multi-stage review protocol. An initial calibration phase was conducted using a sample dataset to harmonise interpretation and identify edge cases. Each labelled sequence then underwent a two-step quality control process: a technical review by the labelling team, followed by an expert review by EUROCONTROL to ensure final quality. In total, 4,536 hours of labelling and 431 hours of reviewing were performed.

### 4.2 Dataset Description

The GVCCS dataset (Jarry et al., 2025) is the first open-access, instance-level annotated video dataset designed for contrail detection, segmentation, and tracking from visible ground-based sky camera imagery. It consists of 122 high-resolution video sequences (totaling 24,228 images) captured at the EUROCONTROL Innovation Hub in Brétigny-sur-Orge, France, using

**Table 1.** Descriptive statistics of the annotated contrail dataset

| Metric | Value |
| --- | --- |
| Total sequences (labelled) | 122 |
| Total images | 24,228 |
| Average sequence duration (minutes) | 96.6 |
| Images per sequence (min / max / mean) | 41 / 600 / 198.6 |
| Total annotated contrail instances | 4,651 |
| Total unique flight IDs assigned | 3,346 |
| Total polygons annotated | 176,234 |
| Contrail duration in minutes (min / max / mean) | 0.5 / 142.5 / 14.6 |
| Polygons per contrail (min / max / mean) | 1 / 589 / 37.8 |
| Polygons per frame per contrail (min / max / mean) | 1 / 4.5 / 1.2 |

Réuniwatt's CamVision sensor. Each sequence has been carefully annotated with temporally consistent polygon masks for visible contrails, including multi-instance tracking and, where possible, attribution to specific flights using aircraft trajectory data.

In total, the annotation team labelled 4,651 individual contrails with a total of 176,194 polygons. The sequences cover a wide range of durations (from 0.5 to 142.5 minutes per contrail), with each contrail comprising between 1 and 589 polygons (mean: 37.8). On average, each video sequence spans 96.6 minutes and contains approximately 193 annotated images. About 3,346 contrails are associated with unique flight identifiers derived from synchronized flight trajectory data filtered above 15,000 ft.

The GVCCS dataset is structured into `train/` and `test/` folders, each containing `images`, `annotations.json` (COCO format), and associated flight data in `parquet` format. The dataset supports a range of research tasks including semantic and panoptic segmentation, temporal tracking, lifecycle analysis, and contrail–flight attribution, and is released under the CC BY 4.0 license.

## 5   Segmentation Models

This section reviews the segmentation models evaluated for identifying, and for some also tracking, contrails. As established in Section 2.2, our primary objective is instance segmentation (detecting individual contrails and assigning them unique identifiers) which is essential for temporal tracking and flight attribution. The models presented here are capable of panoptic segmentation (jointly handling instance identification and scene classification), though our evaluation focuses primarily on contrail instance quality rather than exhaustive scene parsing.

We focus on two model families: Mask2Former, a state-of-the-art transformer-based segmentation model, and a U-Net using a discriminative embedding loss. Both are evaluated on individual images, while only Mask2Former is additionally evaluated on videos.

We also explore two problem formulations: in the single-polygon case, each visible contrail fragment is treated as an independent instance; in the multi-polygon case, all fragments of a given contrail are labelled as a single instance, even if they are spatially disconnected. The single-polygon setting assumes that a subsequent linking algorithm, not implemented in this work, could later group fragments into full contrails. The multi-polygon formulation, in contrast, expects the model to infer such groupings implicitly.

## 5.1 Mask2Former

Mask2Former is a universal segmentation architecture that unifies semantic, instance, and panoptic segmentation within a single model. It is built around a hierarchical encoder-decoder structure comprising three main components: a convolutional backbone for multi-scale feature extraction, a pixel decoder that generates dense spatial embeddings, and a transformer decoder with learnable mask queries that iteratively refines segmentation predictions.

A central innovation in Mask2Former is its use of masked attention in the transformer decoder. Unlike standard cross-attention, which considers the entire image, masked attention limits attention to regions surrounding the current predicted masks. This localized focus enables more precise refinement of object boundaries, which is particularly beneficial for thin, high-aspect-ratio structures like contrails. The model's learnable queries act as object proposals and are refined through multiple decoding layers to generate final instance masks and class labels in an end-to-end manner.

An important aspect of Mask2Former's effectiveness lies in its loss function — the mathematical objective that the model seeks to minimize during training. A loss function quantifies the difference between predicted outputs (e.g., segmentation masks) and ground truth annotations, providing the learning signal that guides iterative parameter updates. The loss function used by Mask2Former combines several components. First, it uses a classification loss that helps the model assign the correct class to each predicted mask (e.g., contrail vs. sky). Second, it includes a mask loss, which measures how closely the predicted
mask matches the ground-truth mask for that object, commonly using pixel-wise binary cross-entropy or Dice loss. Finally, Mask2Former incorporates a matching step based on the Hungarian algorithm (Kuhn, 1955)—a combinatorial optimization method that solves the assignment problem by finding the optimal one-to-one correspondence between two sets given a cost matrix. In this context, the algorithm matches each predicted mask with its most appropriate ground-truth object by minimizing a combined cost based on classification and mask similarity. This optimal matching ensures that each prediction is evaluated
against the correct reference, avoiding duplicate or ambiguous assignments, which is particularly important when multiple contrails with similar appearance are present in the same image.

A detailed technical description of the model is beyond the scope of this paper, as our focus is on applying Mask2Former to contrail segmentation; we refer the reader to the original work by Cheng et al. (2022) for a comprehensive overview of the architecture and performance on popular datasets.

To capture temporal dynamics inherent in contrail evolution, we extend Mask2Former to process short video sequences. Although designed for single images, the model can handle multiple consecutive frames as a 3D spatio-temporal volume by treating time as an additional axis alongside spatial dimensions, following the extension introduced by Cheng et al. (2021a).

Compared to traditional segmentation models, Mask2Former offers substantial architectural advantages. Mask R-CNN (He et al., 2017), while effective, performs detection and segmentation as separate stages, which can introduce spatial misalignment and inefficiencies, especially when segmenting long, disconnected objects. DETR (DEtection TRansformer) (Carion et al., 2020), though end-to-end and transformer-based, primarily focuses on object detection and lacks the fine-grained spatial modelling needed for precise mask prediction. MaskFormer (Cheng et al., 2021b) introduces transformer-based decoding for segmentation but relies on global attention, which can dilute spatial precision. Mask2Former refines this approach with masked attention and iterative refinement, leading to improved accuracy, especially in challenging tasks where objects are often thin, faint, and visually ambiguous.

## 5.2 U-Net with Discriminative Loss

As a baseline, we implement a two-step instance segmentation model. First, we use a U-Net architecture (Ronneberger et al., 2015) for segmentation. U-Net is a convolutional neural network originally developed for biomedical image segmentation, characterized by its distinctive U-shaped architecture. The network features a symmetrical encoder-decoder structure: the encoder progressively downsamples the input to capture high-level semantic features, while the decoder upsamples to recover spatial resolution. Crucially, U-Net employs skip connections—direct pathways that link corresponding encoder and decoder layers, bypassing intermediate processing. These connections allow fine-grained spatial details (such as exact contrail boundaries) that are lost during downsampling to be directly recovered in the decoder, improving the quality and precision of segmentation outputs.

Second, we use a similar architecture that learns a unique feature representation, or embedding, for each pixel in an image by using a discriminative loss function — a training objective specifically designed to encourage pixels from the same instance to have similar embeddings while pushing apart embeddings from different instances. In this model, the final head of the U-Net does not produce a typical segmentation map with class labels. Instead, it produces an embedding for each pixel (a vector in a high-dimensional feature space). The goal is for pixels that belong to the same object instance to have similar embeddings (meaning they are close together in this feature space), while pixels belonging to different instances have embeddings that are far apart. This way, the model effectively learns to group pixels based on their learned features.

The process of identifying individual instances is performed in two separate steps. The first step is to generate these pixel embeddings with the U-Net, and the second step is to group or cluster these embeddings into individual instances. For clustering, we use HDBSCAN (Hierarchical Density-Based Spatial Clustering of Applications with Noise) (Campello et al., 2013)—a density-based clustering algorithm that automatically identifies clusters of arbitrary shape without requiring a predetermined number of clusters. HDBSCAN groups pixels with similar embeddings (high local density in the embedding space) into the same instance while identifying outliers that do not belong to any clear cluster. These outliers are subsequently assigned to the nearest cluster using $k$-means, ensuring complete instance coverage. This approach is particularly suitable for contrails, which

often exhibit irregular, fragmented, or elongated shapes that are difficult to cluster using traditional methods like $k$-means alone.

The discriminative loss function used to train the model is composed of three parts. The first part, known as the pull term, encourages embeddings of pixels that belong to the same instance to be close together, making the cluster compact. The second part, called the push term, forces embeddings of different instances to be sufficiently separated from each other, preventing clusters from overlapping. The third part is a regularization term that prevents the embeddings from growing too large in magnitude, which stabilizes the training process and embedding space. This combination allows the model to learn meaningful and well-separated pixel embeddings without relying on explicit object bounding boxes or pre-defined region proposals. For readers interested in the mathematical formulation and detailed rationale behind the discriminative loss, we refer to the original paper by De Brabandere et al. (2017).

It is important to note that this model operates only on single images. Unlike models such as Mask2Former for videos mentioned in the previous section, it does not incorporate any temporal or sequential information, nor does it include recurrent layers or mechanisms to handle videos. Extending this approach to process video sequences and incorporate temporal consistency would require significant changes to both the architecture and the algorithms used, which is outside the scope of this work.

The embedding-based approach is well suited to segmenting objects that may not be spatially continuous, such as contrails with fragmented shapes. Since the model does not require spatial continuity, it can learn to embed separate, disconnected parts of the same contrail into a similar region of the feature space if they share common visual characteristics and belong to the same label. However, this approach has its challenges. If parts of the same contrail differ significantly in appearance due to factors like changes in lighting, atmospheric conditions, or variations in the background texture, they may be embedded differently and incorrectly assigned to separate clusters. Conversely, visually similar but unrelated contrail fragments could be mistakenly grouped together, as the model relies solely on the learned embeddings for clustering.

Figure 4 illustrates how the discriminative embedding approach learns to separate contrail instances. On the left, the ground truth labels are displayed, highlighting the pixel-wise assignment to contrail instances. On the right, we show the corresponding discriminative embedding space. Since each pixel is represented by a high-dimensional embedding vector (typically 32 dimensions), we apply Principal Component Analysis (PCA) to reduce this to two dimensions for visualization: PCA identifies the two orthogonal directions that capture the most variance in the embedding space, effectively projecting the high-dimensional clusters onto a 2D plane. Each point in this plot represents a single pixel, colored according to its ground-truth instance label. This visualization provides insight into how the model, trained with a discriminative loss, learns to embed pixels from the same instance close together in the feature space, while separating those from different instances. The separation observed in the embedding space confirms the model's ability to cluster fragmented contrail structures, although visually similar but unrelated segments may still partially overlap in the embedding due to shared appearance features.

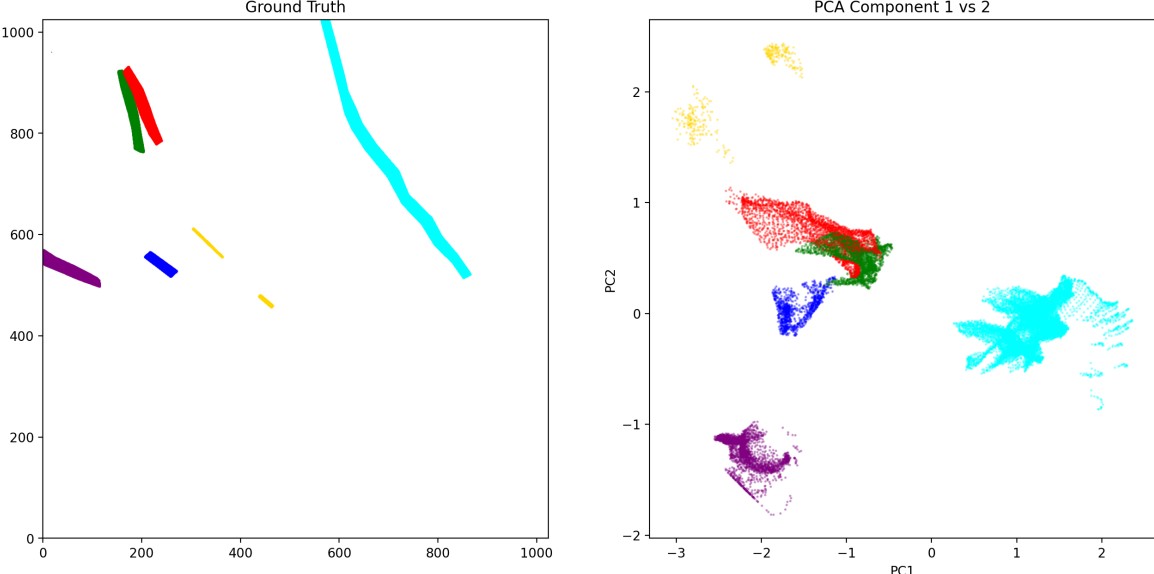

**Figure 4.** Illustration of the discriminative embedding method for instance segmentation. Left panel: Ground-truth (human-annotated) instance labels, where each color represents a distinct contrail. Right panel: Visualization of the learned pixel embeddings. The U-Net model learns to map each pixel to a point in a high-dimensional feature space such that pixels belonging to the same contrail are positioned close together, while pixels from different contrails are far apart. For visualization, PCA reduces this high-dimensional space to two dimensions by identifying the directions of maximum variance. Each point represents one pixel, positioned according to its learned embedding and colored by its ground-truth contrail instance. Well-separated, compact clusters indicate that the model has successfully learned to group pixels from the same contrail while distinguishing different contrails.

## 6 Results

This section presents the performance of the models introduced in Section 5 on contrail segmentation tasks. Our primary goal is not to achieve state-of-the-art results but to establish clear application examples and meaningful baseline performances. By doing so, we highlight the unique opportunities offered by this dataset and provide a foundation for the research community to build upon, encouraging rapid progress in the critical field of aviation's climate impact.

### 6.1 Training

All models were initialized from existing pretrained checkpoints. We trained two versions of the Mask2Former architecture for the single-image segmentation task. Both models share the same core architecture but differ in the size of their transformer backbone: one uses the Swin-Base (Swin-B) configuration and the other uses the larger Swin-Large (Swin-L). The main difference between these two lies in model capacity: Swin-L has significantly more parameters, enabling it to learn richer representations at the cost of higher computational requirements.

Both image models were initialized from publicly available pretrained checkpoints in the Mask2Former Model Zoo[1]. Each model was first pretrained on the ImageNet-21k (IN21k) (Ridnik et al., 2021) classification dataset and then fine-tuned on the COCO panoptic segmentation dataset. While COCO (Lin et al., 2014) does not include contrails, it spans a wide range of natural (including clouds and sky) and man-made objects, offering useful general-purpose segmentation features. This two-stage pretraining (IN21k followed by COCO) has been widely validated in the literature and provides a strong initialization for fine-tuning on contrail imagery.

Both the Swin-B and Swin-L variants were trained on individual image frames using 200 learnable object queries. Given our hardware setup—two NVIDIA RTX 6000 GPUs, each with 48 GB of memory—we were able to train both variants on the image dataset without significant memory limitations.

For video segmentation, we used the video-specific variant of Mask2Former, which extends the original architecture to handle temporal sequences. Like the image-based model, it uses 200 object queries and Swin Transformer backbones, and is initialized from a checkpoint pretrained on the YouTubeVIS 2019 dataset (Yang et al., 2019). Although YouTubeVIS does not contain contrails, its emphasis on learning temporally consistent object masks across frames makes it well suited to capture the dynamics of contrails in video data.

Due to GPU memory constraints, we limited both training and inference to short video clips composed of a small number of consecutive frames. While this restriction was necessary to fit within available hardware resources, particularly for memory-intensive architectures, it also shaped our training strategy. During training, these clips are randomly sampled from longer video sequences to introduce temporal diversity. By varying the starting points of the sampled clips, the model is exposed to contrails at different stages of their lifecycle (formation, elongation, dissipation) and in diverse atmospheric contexts. This stochastic sampling encourages the model to learn more generalizable temporal representations.

To support this setup, we trained the video Mask2Former model using both Swin-Base (Swin-B) and Swin-Large (Swin-L) backbones. However, the number of frames per clip had to be adjusted based on model capacity and memory availability. With the more lightweight Swin-B variant, we were able to train on 5-frame clips, while the higher-capacity Swin-L model could only be trained on 3-frame clips due to its significantly larger memory footprint. This reflects a trade-off between temporal context and model expressiveness: longer clips may better capture the dynamic evolution of contrails, whereas larger models like Swin-L provide richer per-frame representations. Training both configurations allows us to explore how these two dimensions (temporal depth and model capacity) interact in the context of contrail segmentation.

For the U-Net model, we used a backbone based on MaxViT-B, a hybrid vision transformer architecture that combines convolutional layers with self-attention mechanisms for efficient and scalable visual representation learning. This backbone was pretrained on ImageNet-21k and subsequently fine-tuned on ImageNet-1k, providing robust feature representations to support the discriminative loss function employed during contrail segmentation training.

The training procedure for each model involved several epochs of supervised learning, with early stopping applied based on performance on a validation set. The dataset was partitioned into training, validation, and test sets using a 70-10-20 random split at the video level. This means that all frames from a given video were assigned exclusively to one of the three sets to avoid

---

[1]https://github.com/facebookresearch/Mask2Former/blob/main/MODEL_ZOO.md

**Table 2.** Default hyper-parameters for Mask2Former models.

| Hyper-parameter | Default value | Notes / Differences |
| --- | --- | --- |
| Training iterations | 20K | Same for image and video |
| Learning Rate | — | 3.75e-5 (Image), 1.25e-5 (Video) |
| Batch Size | — | 6 (Image), 2 (Video) |
| Image Size | 1024 × 1024 | Same for image and video |
| Class Weight | 2.0 | Same for image and video |
| Mask Weight | 5.0 | Same for image and video |
| Dice Weight | 5.0 | Same for image and video |
| Importance Sample Ratio | 0.75 | Same for image and video |
| Oversample Ratio | 3.0 | Same for image and video |
| Augmentations | Rotation (90°), vertical flip, horizontal flip | Applied at image level (Image); applied at clip level (Video) |

any potential data leakage. To ensure fair and unbiased evaluation, we also balanced the number of empty sequences — videos containing no contrails — across the three subsets.

It is important to note that the reported metrics reflect model performance on contrail-rich scenarios, as the dataset construction deliberately oversampled contrail-positive sequences to maximize training signal. While this choice enhances the dataset's utility for contrail detection and tracking tasks, generalization to unfiltered operational data with arbitrary sky conditions may differ and warrants further investigation.

We did not perform exhaustive hyper-parameter tuning for any of the models. Instead, our goal with this experimental setup was to establish baseline results and to analyze model performance both qualitatively and quantitatively under realistic computational and data constraints. All models were trained using the default hyper-parameters reported in their original publications. Tables 2 and 3 summarize the most important training parameters for each model. Note that the models differ in the specific hyper-parameters relevant to their architecture and training setup. Future work will focus on exploring more sophisticated modeling strategies, systematic hyper-parameter optimization, and additional training refinements.

Each model was trained and evaluated on two distinct formulations of the instance segmentation task. The first formulation treats a contrail as a single object, even if it is composed of multiple disconnected regions or fragmented segments. In this setup, the model must learn to group visually and spatially separated regions that correspond to the same physical contrail. The second formulation simplifies the problem by treating each visible polygon as an independent instance. In this formulation, the model is not required to group disjoint segments belonging to the same contrail; instead, it simply detects and segments each distinct region. This approach corresponds to a modular processing pipeline where instance merging and flight attribution occur at a later stage, as will be discussed in future work.

**Table 3.** Default hyper-parameters for U-Net model trained with discriminative loss.

| Hyper-parameter | Default value |
| --- | --- |
| Architecture | U-Net |
| Backbone | tu-maxvit_base_tf_512.in1k |
| Input image size | $1024 \times 1024$ |
| Precision | 16-mixed |
| Epochs | 100 |
| Batch size | 1 |
| Gradient accumulation steps | 32 |
| Learning rate | $5 \times 10^{-6}$ |
| Optimizer | AdamW (weight decay $= 10^{-4}$) |
| Scheduler | Cosine with warm-up |
| Augmentations | Rotation (90°), vertical flip, horizontal flip |

## 6.2 Evaluation

We evaluate both semantic and instance-level segmentation performance using a combination of standard and task-adapted metrics.

For semantic segmentation, we report the mean Intersection over Union (mIoU) and the Dice coefficient. Both metrics quantify the overlap between predicted and ground-truth masks, with values ranging from 0 (no overlap) to 1 (perfect match). The mIoU is calculated as:

$$mIoU = \frac{\text{Area of Intersection}}{\text{Area of Union}},$$

where the intersection is the set of pixels correctly predicted as contrail, and the union includes all pixels predicted as contrail plus all true contrail pixels. This metric equally penalizes both false positives (predicting contrail where there is none) and false negatives (missing actual contrail pixels).

The Dice coefficient is calculated as:

$$Dice = \frac{2 \times \text{Area of Intersection}}{\text{Size of Prediction} + \text{Size of Ground Truth}}.$$

The factor of 2 in the numerator makes the Dice coefficient emphasize correct overlap more strongly than mIoU. It is particularly sensitive to small or thin structures, making it well-suited for evaluating contrails, which often appear as narrow, elongated features that occupy a small fraction of the image.

## Temporal Evaluation Strategy

For video-based models, inference is performed using a sliding window approach, where each video is divided into overlapping short clips of fixed length, matching the clip length used during training (e.g., 3 frames for the Swin-L model, 5 frames for the Swin-B model). These clips advance by one frame at a time (stride one), allowing the model to leverage temporal context effectively while respecting memory constraints during inference. Crucially, segmentation accuracy is computed only on the central frame of each short clip. This design ensures that each frame in the video contributes exactly once to the evaluation metrics, only when it appears as the center frame of a clip. This prevents duplicate evaluation and enables fair comparison with image-based models, which predict on single frames independently. For example, if a 5-frame clip is used on a video with frames numbered 1 through 10, the first evaluation clip spans frames 1–5 with evaluation on frame 3; the next clip covers frames 2–6 (evaluated on frame 4), and so on. This guarantees unique evaluation for frames 3 to 8, each exactly once.

It should be noted that the video-based Mask2Former model maintains temporally consistent instance identifiers within each clip. That is, if a contrail is labelled as instance #3 in one frame of a clip, it retains this identifier across all frames in the same clip. However, since clips are processed independently, these identifiers are not guaranteed to remain consistent between consecutive clips. A given contrail may receive a different identifier in adjacent clips. To enable continuous tracking of contrails throughout the entire video, we introduce a simple post-processing method that links and reconciles these instance identifiers to generate coherent, continuous tracks; this method is described in detail in A.

## Instance Segmentation Metrics

Model performance is evaluated using both semantic and instance-level segmentation metrics. All metrics are computed globally by aggregating predictions and ground truths across the entire test set before applying the metric calculations. This global computation prevents biases that can arise from averaging metrics computed independently on each observation (i.e., frame), which is particularly important in settings with imbalanced or sparse data such as contrail segmentation.

Instance segmentation performance is assessed using COCO-style metrics (Lin et al., 2014) computed globally over the dataset. To accommodate the specific challenges posed by contrails, we adapt the IoU threshold range. The notation X@[IoU range | size category | max detections] specifies three parameters:

– IoU range: The range of Intersection over Union thresholds used. A prediction is considered a "true positive" only if its IoU with a ground-truth object exceeds the threshold. Average Precision (AP) is computed across multiple thresholds and averaged.

– Size category: Filters objects by area —"small" ($< 32^2$ pixels), "medium" ($32^2$ to $96^2$ pixels), "large" ($> 96^2$ pixels), or "all" (no filtering).

– Max detections: The maximum number of predicted instances considered per image (e.g., 100).

**Table 4.** Semantic segmentation metrics. For the Mask2Former variants, values without parentheses refer to Swin-B; values in parentheses refer to Swin-L.

| | Single Images | | Videos |
|---|---|---|---|
| **Metric** | **Mask2Former** | **U-Net** | **Mask2Former** |
| Dice | 0.56 (0.60) | 0.59 | 0.57 (0.59) |
| mIoU | 0.38 (0.43) | 0.42 | 0.40 (0.42) |

For example, AP@[0.25:0.75 | all | 100] denotes Average Precision computed over IoU thresholds ranging from 0.25 to 0.75, across all object sizes, with a maximum of 100 detections evaluated per image. In the results that follow, we report both Average Precision (AP) and Average Recall (AR) using this notation.

We restrict the IoU threshold range to [0.25, 0.75], rather than the standard COCO range of [0.50, 0.95], to better accommodate the elongated and thin geometry of contrails, where very high IoU thresholds are overly strict. Contrails are thin, irregular, and may extend across large image portions, making exact mask overlap challenging. A prediction overlapping 30% of a contrail would be ignored under COCO's default minimum IoU of 0.5 but counted as a true positive under our more lenient thresholds. This adaptation better reflects practical segmentation quality for contrails.

By adjusting the IoU range, the metrics better reflect practical segmentation quality for contrails, balancing sensitivity to spatial accuracy with tolerance for slight misalignments and fragmentations inherent to this domain. It is important to note that these adapted metrics are not directly comparable to standard COCO scores but are specifically tailored to provide meaningful evaluation in the context of contrail segmentation.

This evaluation framework, combining semantic and instance segmentation metrics computed globally with appropriate IoU thresholds and size categories, offers a comprehensive and interpretable means of assessing model performance. It facilitates fair comparisons across models and supports future benchmarking on our contrail dataset.

Tables 4 and 5 summarize the results for the semantic and instance segmentation tasks, respectively. All results are reported for both single-image and video-based models. Instance segmentation results are further disaggregated by annotation style: **M** refers to multi-polygon annotations, and **S** refers to single-polygon annotations. For Mask2Former models, values without parentheses correspond to the Swin-B backbone, while those in parentheses refer to Swin-L.

In the semantic segmentation task, performance remains consistent across all models and variants, with Dice and mIoU scores showing little variation. This stability is expected, as semantic segmentation only requires classifying each pixel as either contrail or sky, without distinguishing between separate contrail instances. The U-Net model achieves results on par with the more advanced Mask2Former models, indicating that per-pixel contrail detection is largely driven by local visual features, such as shape, brightness, and texture, which U-Net captures effectively.

These results also reflect the quality and consistency of our dataset: although based on ground-level imagery, the segmentation performance is in line with results reported in previous studies using satellite data (Jarry et al., 2024; Ortiz et al., 2025).

**Table 5.** Instance segmentation metrics. "M" refers to multi-polygon, whereas "S" indicates single-polygon. For the Mask2Former variants, values without parentheses refer to Swin-B; values in parentheses refer to Swin-L.

| Type | Metric | Single Images | | Videos |
| | | Mask2Former | U-Net | Mask2Former |
| --- | --- | --- | --- | --- |
| M | AP@[0.25:0.75 \| all \| 100] | 0.34 (0.34) | 0.05 | 0.31 (0.33) |
| | AP@[0.25:0.75 \| small \| 100] | 0.21 (0.21) | 0.01 | 0.14 (0.17) |
| | AP@[0.25:0.75 \| medium \| 100] | 0.39 (0.40) | 0.13 | 0.37 (0.38) |
| | AP@[0.25:0.75 \| large \| 100] | 0.44 (0.47) | 0.12 | 0.46 (0.47) |
| | AR@[0.25:0.75 \| all \| 1] | 0.10 (0.10) | 0.03 | 0.09 (0.09) |
| | AR@[0.25:0.75 \| all \| 10] | 0.41 (0.41) | 0.18 | 0.38 (0.40) |
| | AR@[0.25:0.75 \| all \| 100] | 0.44 (0.44) | 0.22 | 0.43 (0.44) |
| | AR@[0.25:0.75 \| small \| 100] | 0.30 (0.30) | 0.14 | 0.26 (0.29) |
| | AR@[0.25:0.75 \| medium \| 100] | 0.50 (0.50) | 0.25 | 0.49 (0.50) |
| | AR@[0.25:0.75 \| large \| 100] | 0.55 (0.55) | 0.22 | 0.57 (0.56) |
| S | AP@[0.25:0.75 \| all \| 100] | 0.35 (0.37) | 0.06 | 0.31 (0.34) |
| | AP@[0.25:0.75 \| small \| 100] | 0.24 (0.26) | 0.03 | 0.17 (0.21) |
| | AP@[0.25:0.75 \| medium \| 100] | 0.44 (0.45) | 0.14 | 0.41 (0.43) |
| | AP@[0.25:0.75 \| large \| 100] | 0.37 (0.43) | 0.11 | 0.46 (0.47) |
| | AR@[0.25:0.75 \| all \| 1] | 0.08 (0.08) | 0.03 | 0.07 (0.08) |
| | AR@[0.25:0.75 \| all \| 10] | 0.37 (0.38) | 0.18 | 0.35 (0.37) |
| | AR@[0.25:0.75 \| all \| 100] | 0.44 (0.45) | 0.21 | 0.42 (0.45) |
| | AR@[0.25:0.75 \| small \| 100] | 0.33 (0.34) | 0.15 | 0.28 (0.32) |
| | AR@[0.25:0.75 \| medium \| 100] | 0.53 (0.53) | 0.26 | 0.52 (0.55) |
| | AR@[0.25:0.75 \| large \| 100] | 0.54 (0.56) | 0.25 | 0.58 (0.60) |

Although differences in imaging modality and scene geometry preclude direct comparisons, the consistency in results suggests that semantic contrail segmentation is a well-posed task for modern architectures, with strong performance achievable across diverse data sources.

Instance segmentation results reveal clear differences between model architectures. These differences are more substantial than those observed in the semantic segmentation task, highlighting the added complexity introduced by instance-level reasoning. Mask2Former, which is designed for panoptic segmentation through object-level queries and global spatial reasoning, consistently outperforms U-Net across all instance metrics. The performance gap is particularly pronounced in the multi-polygon setting, where contrails appear fragmented and must be correctly grouped into coherent instances. These results

highlight the value of architectures specifically built for instance-aware tasks: Mask2Former's ability to reason globally and associate disjoint segments makes it better suited for detecting and tracking individual contrails.

A more nuanced comparison emerges when evaluating image-based versus video-based Mask2Former models. For the Swin-B backbone, the image-based model achieves higher instance segmentation performance, while the video-based model slightly outperforms it on semantic segmentation metrics. This suggests that although video models benefit from temporal consistency and motion cues, the added complexity of enforcing cross-frame coherence may introduce challenges that slightly hinder instance-level prediction accuracy, particularly when using a lower-capacity backbone like Swin-B.

In the Swin-L setting, the image-based model performs best overall. It achieves both the highest instance segmentation score and slightly superior semantic segmentation performance. These results indicate that temporal modeling does not always yield performance improvements, especially when the temporal context is limited (e.g., 3-frame clips) or when the spatial representation capacity of the model is already high. The image-based model benefits from pretraining on COCO, which may favor precise spatial delineation, while the video-based variant relies on pretraining on YouTubeVIS, which is more focused on temporal coherence. However, it is important to note that the video-based model performs an additional task: tracking. By maintaining consistent instance identities across frames, it enables temporally coherent segmentation that is not achievable with image-based models. The metrics reported here are computed on a per-frame basis and do not account for flickering or instance identity consistency over time. These temporal aspects are particularly important in video applications and are not captured by the conventional frame-level evaluation scores presented herein.

An important caveat is that all reported metrics are computed independently for each frame and do not account for temporal consistency of instance identities over time. Video-based models are explicitly trained to maintain coherent instance tracks across frames through end-to-end temporal modeling, jointly optimizing segmentation and tracking within a unified objective. In contrast, image-based models require post-hoc association algorithms (such as the Hungarian matching method described in A) to link instances temporally based on spatial overlap alone. While both approaches can achieve tracking, video models learn temporal correspondences from motion cues and appearance features during training, potentially offering more robust handling of occlusions, fragmentations, and brief disappearances. However, the per-frame metrics reported here (AP, AR, Dice, mIoU) primarily assess spatial segmentation quality and do not reward temporal consistency. As a result, while video models do not uniformly outperform image models in per-frame scores, they provide qualitative benefits in terms of reduced instance ID flickering and smoother temporal transitions that are not captured by these metrics. Future work should incorporate video-specific evaluation metrics (e.g., tracking accuracy, ID switches, fragmentation) to fully characterize the advantages of temporal modeling. Additionally, the short clip lengths used in this study (3–5 frames) were dictated by hardware constraints; longer temporal contexts may yield further improvements and warrant investigation with more capable architectures.

Overall, Swin-L outperforms Swin-B across all setups, reinforcing the benefit of increased model capacity for fine-grained spatial understanding and instance-level reasoning. Nonetheless, this comes at the cost of higher computational requirements, particularly in the video setting, underscoring a trade-off between performance and scalability.

Another important trend observed in the evaluation is that model performance is strongly influenced by contrail size and detection caps. Generally speaking, larger contrails are segmented more accurately due to their higher pixel counts and lower am-

biguity, while allowing more predicted instances (e.g., increasing the detection limit) improves recall by removing constraints on how many objects can be reported. These trends are consistent with general findings in object detection and reinforce the shared challenges between contrail segmentation and broader instance segmentation tasks.

Comparing the multi-polygon and single-polygon formulations reveals a difference in task difficulty: the single-polygon setting is inherently easier. Across all models and data modalities, instance segmentation metrics are consistently higher when using the single-polygon formulation. This is because the task removes the need to group fragmented or spatially disjoint contrail segments into separate instances. Instead, all parts of a contrail, regardless of their separation, are treated as a single mask, greatly simplifying the model's objective. The model is no longer required to learn complex grouping strategies or reason over spatial and temporal discontinuities. Note that semantic segmentation metrics remain virtually unchanged between the two formulations, indicating that identifying contrail pixels is equally feasible in both cases. The difference lies solely in how those pixels are grouped into instances. This distinction confirms that the main challenge in the multi-polygon task is not pixel classification but instance association.

These results have important practical implications for different contrail detection scenarios. For older contrails, such as those typically observed in satellite imagery or in ground-based images when the contrail formed outside the camera's field of view, it is extremely difficult to associate the contrail with its source flight. In these cases, the only viable option is to group visible fragments into instances based solely on visual information. This makes multi-polygon instance segmentation essential, as it allows models to detect and associate disjoint contrail segments without relying on external data. Our dataset and Mask2Former-based models are specifically designed for this setting, enabling effective instance-level detection even when contrails are fragmented, occluded, or spatially disconnected.

In contrast, when a contrail forms directly above the camera and additional data such as aircraft trajectories and wind fields are available, a different approach becomes feasible. In these situations, one can perform single-polygon instance segmentation, where contrail fragments are grouped into a single instance using post-hoc association based on flight paths and advection. This formulation is simpler from a computer vision perspective and is commonly used in the literature (Ortiz et al., 2025; Chevallier et al., 2023; Van Huffel et al., 2025), mainly because multi-polygon annotated datasets have not been available until now. However, this method depends on access to external data and is only applicable to contrails formed during the observation window, after the aircraft has entered the scene.

By supporting both the multi- and single-polygon formulations, our dataset enables training and evaluation across a broader set of operational use cases. The multi-polygon task is essential for vision-only detection of older contrails or those in satellite imagery, while the single-polygon formulation may be more suitable when additional metadata enables contrail-to-flight attribution. This distinction will be further explored in future work focused on linking contrails to their source aircraft.

### 6.3 Illustrative examples

We present two test-set examples to illustrate the challenges of the multi-polygon contrail segmentation task. In both cases, we compare predictions from image-based and video-based versions of the Mask2Former model, trained from pretrained Swin-

L backbones. These examples highlight how temporal context affects instance predictions and expose typical failure modes, including contrail fragmentation, occlusion by clouds, and confusion between contrails and visually similar cloud structures.

Figure 5 shows a frame from April 25, 2024 at 05:51:00 (UTC), under clear-sky conditions. The background is uniformly blue, providing favorable conditions for both human and machine segmentation. The corresponding ground-truth annotations include several contrails labelled as fragmented (e.g., identifiers 0, 1, and 5), based on known flight trajectories available to annotators during the labelling process. This makes the example suitable for evaluating instance-level understanding in the multi-polygon setting.

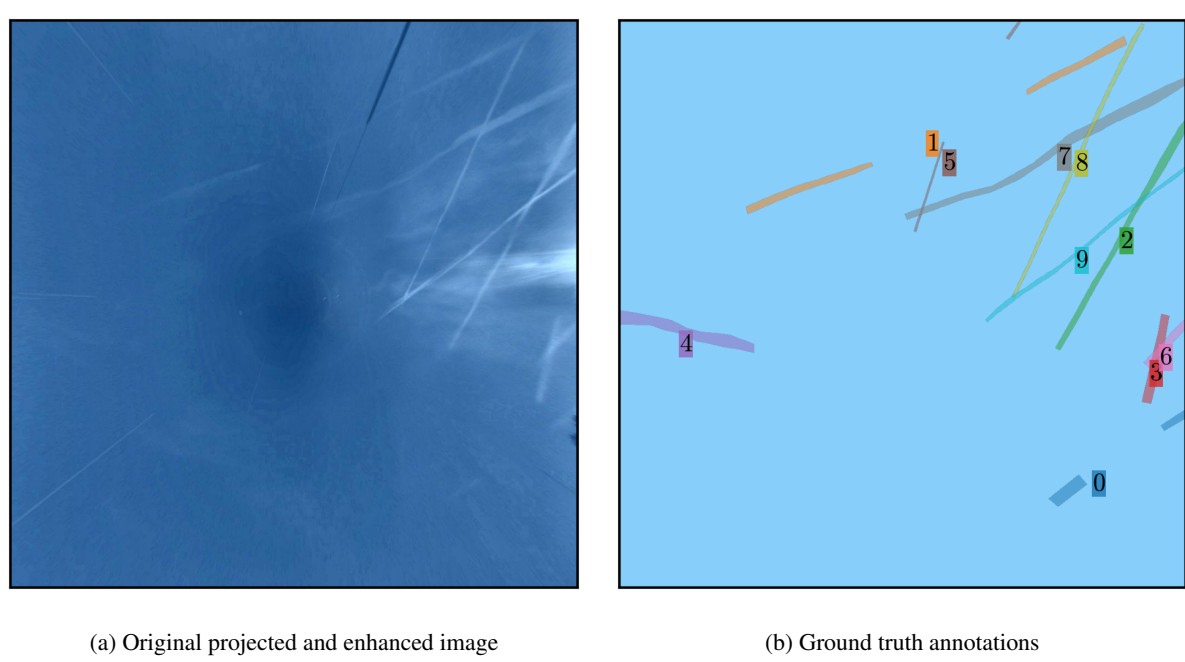

(a) Original projected and enhanced image        (b) Ground truth annotations

**Figure 5.** Original projected and enhanced as well as ground truth annotations for April 25, 2024 at 05:51:00 (UTC).

Figure 6 shows predictions from both models for this scene. Despite the favorable background, both models exhibit instance-level errors. The image-based model correctly infers that contrail 1 is fragmented but detects just one segment of contrail 0, missing the other entirely. It completely misses contrail 4 and erroneously merges contrails 3 and 6 into a single prediction. The video-based model makes similar mistakes: it also merges contrails 3 and 6, and fails to detect contrail 4. Additionally, it predicts the second fragment of contrail 0 but assigns it to a different instance, and it incorrectly splits contrail 1 into two separate instances.

From a semantic segmentation perspective, both models perform relatively well, as expected in a high-contrast scene. The image-based model achieves a Dice score of 0.76 and a mean IoU of 0.64, while the video-based model slightly outperforms it with a Dice of 0.79 and mean IoU of 0.67. However, due to the instance grouping errors, the image model achieves a slightly higher AP@[0.25:0.75 | all | 100] (0.62) than the video model (0.55).

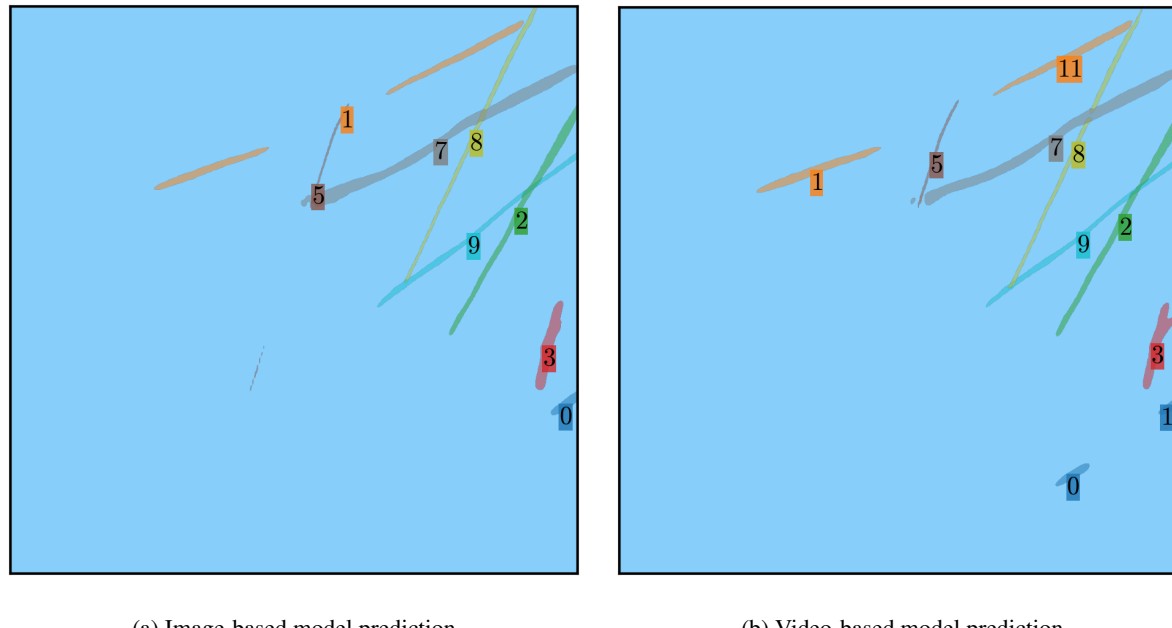

(a) Image-based model prediction                    (b) Video-based model prediction

**Figure 6.** Predicted instances for the frame shown in Fig. 5, using Swin-L models with image and video inputs.

Figure 7 shows a more challenging frame captured on November 19, 2023 at 08:49:30 (UTC). Here, several cirrus clouds are present in the background, which introduces ambiguity, as some of these cloud structures resemble contrails. This scene also includes multiple contrails that are spatially aligned and fragmented, increasing the complexity of the instance segmentation task.

This scene illustrates a common failure mode: fragmentation and misgrouping of visually aligned but semantically distinct contrails. Contrail 6 is split into two segments with contrail 0 lying in between; although they appear collinear, contrail 0 is a distinct instance generated by a separate flight. Contrail 7 appears shortly after and may be misassociated with contrails 6 and 0 in the absence of flight metadata. The image-based model correctly separates contrail 0 from 6 but incorrectly merges contrails 6 and 7. The video model groups all three (6, 0, and 7) into a single prediction. Interestingly, this error reflects a plausible human interpretation without flight context, highlighting the challenge of the task.

Both models fail to detect contrails 1 and 8, which are partially occluded by clouds. They also produce a false positive (labelled as contrail 9), segmenting a cirrus structure that resembles a contrail. While the dataset is of high quality and was carefully annotated with access to flight information, some visually ambiguous cases, such as the one discussed, remain inherently difficult to label with certainty. In this example, the predicted region resembles a contrail in both structure and intensity, making it unclear whether the false positive stems from a model error or an understandable omission in the ground truth. These rare edge cases highlight the potential influence of mild label noise in visually complex scenes. Future work could benefit

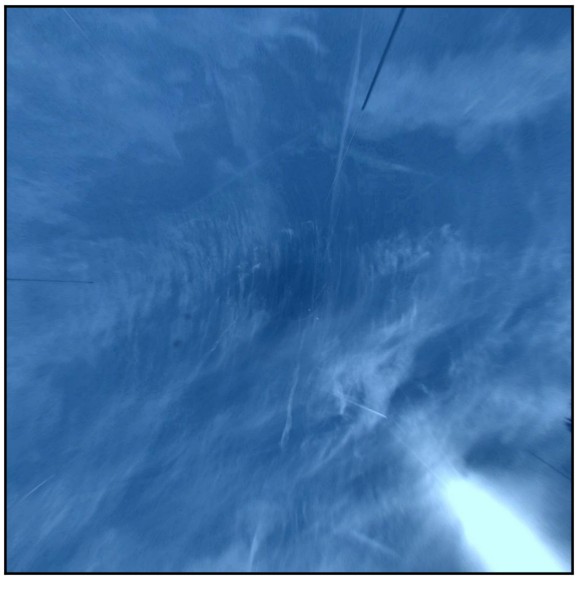 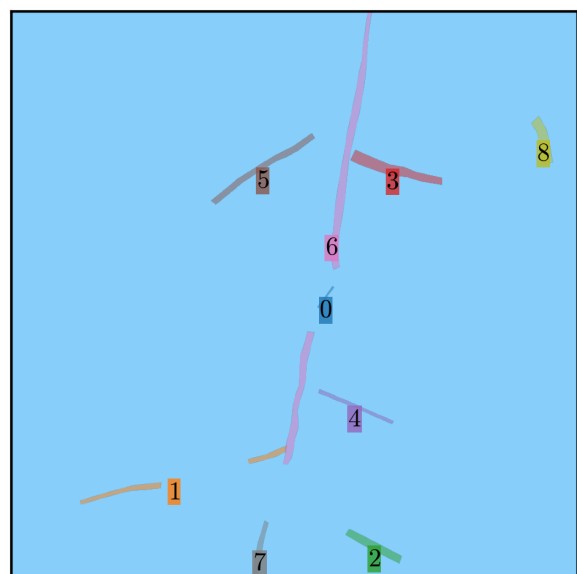

(a) Original projected and enhanced image
(b) Ground truth annotations

**Figure 7.** Original projected and enhanced image as well as ground truth annotations for November 19, 2023 at 08:49:30 (UTC).

from complementary strategies such as confident learning (Northcutt et al., 2021) to further refine annotations and improve robustness in borderline cases.

Semantic segmentation performance in this scene is lower than in the previous one, reflecting increased difficulty. The image model achieves a Dice score of 0.61 and mIoU of 0.43, while the video model scores 0.70 and 0.54, respectively. Instance-level AP@[0.25:0.75 | all | 100] scores are 0.35 and 0.37, respectively, similar to the average metrics, making this a representative case.

These examples illustrate several key challenges in multi-polygon contrail segmentation: (1) correct grouping of fragmented contrail segments from the same flight; (2) visual ambiguity due to clouds that resemble contrails; (3) occlusion; and (4) spatial overlap of contrails from different flights. While video-based models benefit from temporal information, they may over-group distinct instances. Image-based models avoid this but often fail to connect fragmented segments. Overall, these examples demonstrate the inherent difficulty of the task and the limitations of current models.

## 7 Conclusions

This work introduces a new dataset (Jarry et al., 2025) and baseline models for contrail segmentation from ground-based camera imagery. Our experiments show that modern computer vision methods, particularly panoptic segmentation models like Mask2Former, can be effectively applied to this task, especially when using large pretrained models and temporal information.

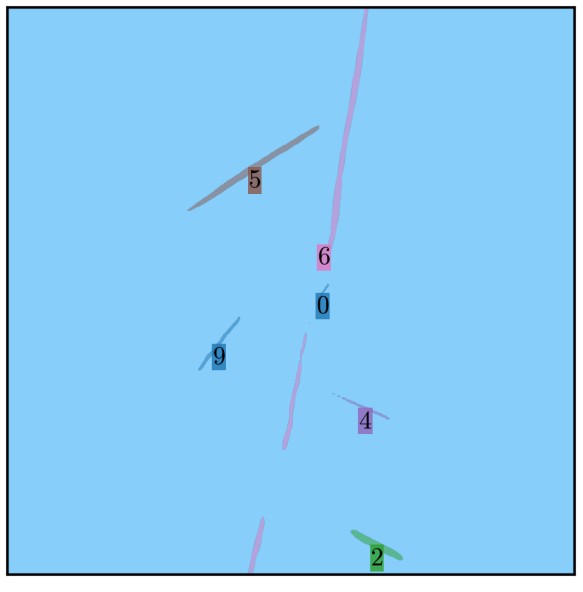 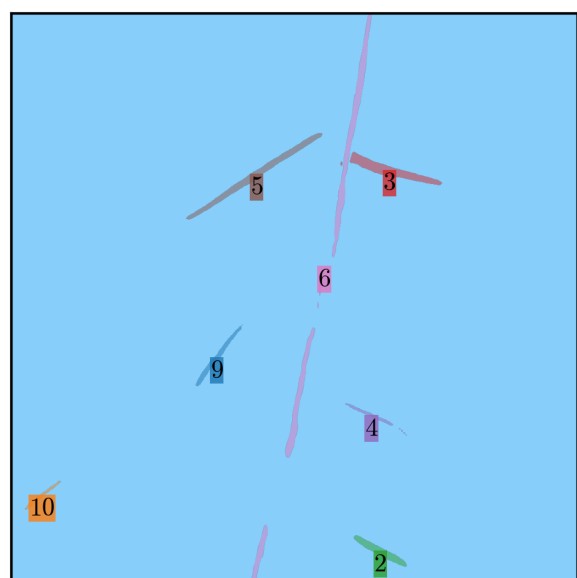

(a) Image-based model prediction           (b) Video-based model prediction

**Figure 8.** Predicted instances for the frame shown in Fig. 7, using Swin-L models with image and video inputs.

However, performance gains often come at the cost of increased computational and memory demands, highlighting a trade-off between accuracy and practicality.

The main contribution of this study is the release of the first video-annotated dataset specifically designed for instance-level contrail segmentation, tracking, and flight attribution in the visual spectrum. Along with detailed evaluation metrics, including average precision and recall across multiple intersection-over-union thresholds and object size bins, this benchmark provides a reproducible baseline for further research in this emerging field.

A key limitation of our current setup is that the visible-light camera restricts observations to daytime conditions. Yet contrails often have their greatest radiative impact at night, when they reduce outgoing longwave radiation and contribute to atmospheric warming. To address this, we are deploying a co-located infrared imaging system that enables continuous, day-and-night monitoring. This may also allow us to begin estimating the radiative forcing of individual contrails under real atmospheric conditions.

In parallel, we are working on a contrail-to-flight attribution algorithm that links observed contrails to specific aircraft using ADS-B trajectory data. This tool, and the associated data and code, will be openly released in a future publication. Attribution is of utmost importance because it allows each contrail to be linked to detailed aircraft and engine parameters, such as aircraft type, engine model, fuel burn rate, flight altitude, and ambient conditions. These inputs are necessary to reproduce the contrail using physical models like CoCiP, assess its expected properties (e.g., ice crystal number, optical depth, lifetime), and ultimately validate or refine these models using real-world observations.

We are also extending this work by annotating a new dataset of contrails in satellite imagery, with instance-level and sequence-based labels. This dataset will allow us to test and evaluate the full multi-scale tracking pipeline proposed in this paper: starting from high-resolution, ground-based detection, followed by attribution to flights, and finally linking to the same contrails as they evolve in satellite imagery. This approach offers a unique opportunity to study contrail formation, spreading, and dissipation over time and at scale. We also plan to use our ground-based dataset to evaluate the predictions of physical models such as CoCiP. Direct comparisons between observed and simulated contrail evolution will help assess model accuracy and potentially inform improvements in contrail forecasting and climate modeling.

Ideally, contrail detection, tracking, and attribution should be addressed by a single deep learning architecture capable of jointly processing video, flight trajectory data, and meteorological fields. For instance, a variant of Mask2Former could be adapted for this purpose. Such an integrated approach would enable end-to-end learning and exploit the complementary nature of the inputs, as weather conditions and aircraft traffic data are highly informative for both detecting and tracking contrails. However, this integration is not straightforward. It requires careful design of input data representations to handle spatio-temporal and multi-modal inputs, the creation of aligned and consistent annotations for all tasks, and the development of loss functions that balance competing objectives across detection, segmentation, tracking, and attribution. Despite these challenges, we encourage the research community to explore this unified approach.

Additionally, deploying multiple cameras in a spatially distributed network would enable stereographic height analysis: contrails observed simultaneously from different viewing angles could be triangulated to determine altitude directly, rather than assuming a fixed height. This would provide crucial validation data for contrail formation models and improve flight attribution accuracy by eliminating altitude uncertainty.

More broadly, we hope this work encourages the development of similar ground-based contrail monitoring systems in other regions. A collaborative, open-science approach — sharing datasets, models, and observational infrastructure — will be essential to building a geographically diverse and temporally continuous picture of contrail behavior. We view this paper as a first step toward a collaborative, open-science framework for contrail research: one that integrates physical modeling with observational data through openly shared datasets and tools, spans spatial and temporal scales through multi-platform monitoring, and supports long-term efforts to better understand and reduce aviation's impact on climate. By providing high-quality ground-based data alongside baseline computer vision models, we aim to facilitate model-data comparison, enable validation of physical models, and encourage the development of complementary monitoring systems worldwide.

## 8 Data availability

The GVCCS dataset is openly available on Zenodo at https://doi.org/10.5281/zenodo.15743988 (Jarry et al., 2025) under a CC BY 4.0 license. The dataset includes 122 video sequences with instance-level annotations, images, COCO-format annotations, and associated flight data in parquet format. The dataset is structured into training and test folders, each containing images, annotations in COCO JSON format, and flight trajectory data.

## Appendix A:  Consistent Instance Tracking Algorithm

Due to memory limitations, the video segmentation model operates on short temporal clips of fixed length $N$ frames, using a sliding window of stride 1. While instance segmentation within each clip is temporally consistent (i.e., instance identifiers are maintained across frames within the clip), the model processes each clip independently. As a result, instance identifiers are not necessarily consistent across clips.

To enforce globally consistent instance identifiers across the full video sequence, we implement a deterministic post-processing method that aligns instance predictions across overlapping clips. The method uses mask overlap similarity—specifically, IoU—across shared frames and performs optimal bipartite matching using the Hungarian algorithm. Below, we provide a rigorous description of the method.

For a given frame index $t \in \{N, N+1, \ldots, T\}$, we define:

- The **current clip** as the sequence $F_{t-N+1}, F_{t-N+2}, \ldots, F_t$.

- The **previous clip** as the sequence $F_{t-N}, F_{t-N+1}, \ldots, F_{t-1}$.

The two clips overlap in $N-1$ frames: $F_{t-N+1}, \ldots, F_{t-1}$. Only frame $F_t$ is newly introduced in the current clip. At each step, we seek to propagate consistent instance identifiers by matching instances across the overlapping frames. Let:

- $\mathcal{I}_{\text{prev}} = \{1, \ldots, K\}$: instance identifiers in the previous clip.

- $\mathcal{I}_{\text{curr}} = \{1, \ldots, M\}$: instance identifiers in the current clip.

We define a cost matrix $C \in \mathbb{R}^{M \times K}$, where each element $C_{ij}$ encodes the negative temporal IoU between instance $i \in \mathcal{I}_{\text{curr}}$ and instance $j \in \mathcal{I}_{\text{prev}}$ over the overlapping frames:

$$C_{ij} = -\frac{1}{N-1} \sum_{f=t-N+1}^{t-1} \text{IoU}\left(\mathcal{M}_{i,f}^{\text{curr}}, \mathcal{M}_{j,f}^{\text{prev}}\right),$$

where $\mathcal{M}_{i,f}^{\text{curr}}$ and $\mathcal{M}_{j,f}^{\text{prev}}$ denote the binary masks of instances $i$ and $j$ at frame $f$, respectively. If an instance does not appear in a given frame (e.g., missing mask), its contribution is treated as zero overlap.

To eliminate unlikely or noisy matches, we apply a threshold $\tau \in [0,1]$ on the mean IoU:

$$C_{ij} = \begin{cases} C_{ij} & \text{if } -C_{ij} \geq \tau, \\ +\infty & \text{otherwise.} \end{cases}$$

where the threshold $\tau$ is selected empirically to balance precision and robustness; we recommend $\tau = 0.1$.

We remove rows and columns of the cost matrix that contain only $+\infty$ entries. Using the modified cost matrix, we solve the bipartite assignment problem via the Hungarian algorithm (Kuhn, 1955)—an optimization method that finds the optimal one-to-one matching minimizing total cost—obtaining a one-to-one (or partial) mapping between current and previous instances.

Let $\sigma : \mathcal{I}_{\text{curr}} \to \mathcal{I}_{\text{prev}} \cup \{\varnothing\}$ denote the resulting assignment. We then update the instance identifiers in the current clip to match those of the assigned instances in the previous clip. Unmatched instances are assigned new unique identifiers. The pseudo-code of the algorithm is presented in Algorithm A1.

---

**Algorithm A1** Post-processing for Consistent Instance Tracking

---

**Require:** Predicted instance masks for video frames $F_1, \ldots, F_T$, threshold $\tau$

1: Initialize unique identifier counter
2: Previous clip instances $\leftarrow$ Predicted instances on clip $(F_1, \ldots, F_N)$
3: Assign unique identifiers to all instances in Previous clip instances
4: **for** $t = N + 1$ to $T$ **do**
5:     Current clip instances $\leftarrow$ Predicted instances on clip $(F_{t-N+1}, \ldots, F_t)$
6:     Compute cost matrix $C$ over frames $F_{t-N+1}, \ldots, F_{t-1}$
7:     Apply threshold $\tau$ and prune rows/columns with all $+\infty$
8:     $\sigma \leftarrow$ Hungarian Algorithm$(C)$
9:     Update instance identifiers in Current clip instances using mapping $\sigma$
10:     Assign new identifiers to unmatched instances
11:     Previous clip instances $\leftarrow$ Current clip instances
12: **end for**

---

This process is applied sequentially from frame $t = N$ to $T$, ensuring that instance identifiers are globally consistent across the video.

*Author contributions.* G.J.: Conceptualization, Data curation, Formal analysis, Investigation, Methodology, Software, Visualization, Writing (original draft preparation). R.D.: Formal analysis, Investigation, Methodology, Software, Visualization, Writing (original draft preparation). P.V.: Conceptualization, Data curation, Formal analysis, Funding acquisition, Investigation, Resources, Software, Writing (original draft preparation). F.B.: Data curation, Project administration, Supervision, Writing (review and editing). S.D.B.: Data curation, Formal analysis, Investigation, Methodology, Software, Writing (review and editing).

*Competing interests.* The authors declare that they have no conflict of interest.

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
