# Peer review of "GVCCS: A Dataset for Contrail Identification and Tracking on Visible Whole Sky Camera Sequences"

_Earth System Science Data, 2025_

## Referee Comment (RC2)

Review of G. Jarry et al.,"GVCCS: A Dataset for Contrail Identification and Tracking on Visible Whole Sky Camera Sequences"

The paper deals with an important task: The setup of an observational database of contrail properties suitable for checking the validity of contrail model results.

The paper describes the dataset "GVCCS: Ground Visible Camera Contrail Sequences repository" which one (including me) could download from https://doi.org/10.5281/zenodo.16419651.

The dataset contains video pictures with very many whole-sky camera figures showing all kinds of clouds including contrails and a set of parquet files which I cannot read because I have no Python knowledge (which tool could one use to open these files and to look at these data???).

The paper gives a technical description of how the files are produced and handled. Also, the paper describes 1. the goals, 2. some of the background knowledge, 3. the state of the art, 4. the datasets, 5. the segmentation models, 6. the results, and 7. the conclusions. - This structure obviously contains some overlaps (e.g., background and state of the art).

The method provides a "segmentation" which means a list of segment elements with their basic properties, such as representing a contrail.

The method identifies boxes containing contrails and "Instance segmentation" which I learned here that this means the pixels containing contrail cloud elements and allows for "panoptic segmentation" which assigns each pixel to a unique contrail. (I must say, I am not quite sure that I have understood this correctly; these terms are new for me).

For me the paper was hard to read and I did by far not understand all details.

I can say that the paper is worth to have and it should be published.

However, I recommend that the authors go through the paper together with somebody who has no or only little experiences in all the tools and terms used, to make sure that all terms used are well defined and to improve the readability.

Specifically, I miss a discussion of several older papers on this and related topics. For example, the paper misses work of the team around Mannstein (- 2013). You cite one of his papers (Mannstein et al., 1999)

but there are several others: (Meyer et al., 2002; Mannstein, 2005; Mannstein and Schumann, 2005; Minnis et al., 2005; Palikonda et al., 2005; Meyer et al., 2007; Zinner et al., 2008; Graf et al., 2009; Schumann et al., 2009; Mannstein et al., 2010; Mannstein et al., 2012; Ewald et al., 2013; Vázquez-Navarro et al., 2013; Vázquez-Navarro et al., 2015). I recommend to check also the following papers for related work: (Bugliaro et al., 2012; Iwabuchi et al., 2012; Hamann et al., 2014; Kox et al., 2014; Strandgren et al., 2017b; Strandgren et al., 2017a).

Some of these references should be included in the Introduction, for example at line 40/41.

Line 106: Appleman did not explain the process completely. For example, the impact of engine efficiency was not understood by him. Schmidt (1941) was the first who understood that contrails form when the ambient temperature is low enough so that the humidity inside the plume gets high enough to let water condense Schumann (1996) covers the concept more completely, including engine efficiency and practical analysis methods, and introduced the term "Schmidt-Appleman Criterium".

Line 119, see also Mannstein et al. (2010).

Line 143: here and at many other places in this paper, the authors refer to technical aspects without citing proper literature. Here: Where can I read more about COCO? (see also Line 328).

Line 145: If you say "traditionally", then there should be lots of examples in the literature. But I miss citations.

Line 151: Instance segmentation was done, e.g. by Mannstein et al. (1999).

How would you classify the methods used by Schumann et al. (2013)?

Line 219: Not only Meijer but others did similar work: E.g., Vázquez-Navarro et al. (2010) firsgt identified contrail segments in high-resolution Low- Earth-Observation data (MODIS) and then followed the contrails in time in a sequence of Meteosat data (with high time resolution). I assume that has been done similarly also by others.

Line 244: what is the meaning of "non-data driven"?

Line 294: what is CLAHE - is there a reference? Line 333: What is a U-Net? Line 350: what is a loss-model? Line 355: what is a Hungarian algorithm? Line 376: what are skip connections? Line 389: what is the HDBSCAN algorithm? Line 410. I have some general knowledge on Principle Component Analysis, but I do not see how this is applied here. Fig. 4, caption: When is a label "true"? What is "discriminative embedding"??? Chapter 6 is indigestible, at least for me. It needs careful revision. Page 23: Where is Fig. 6 mentioned? Line 619: What is a Dice score and what is mIoU, and what is AP@[0.25:0.75] etc. meaning? Line 638. One word seems to be too much in "we are deployed" Line 665: I have no idea what an "data-driven ecosystem" could be. Very minor detail: Line 80: Unnecessary comma after the 5th word.

At the end you might also discuss the idea of using several cameras distributed in sufficient neighborhood such that each contrail is seen by 2 cameras. This would allow for stereographic height analysis.

**References**

- Bugliaro, L., Mannstein, H., and Kox, S.: Ice clouds properties from space, in: Atmospheric Physics Background, Methods, Trends, edited by: Schumann, U., Springer, Heidelberg, doi: 10.1007/978-3-642-30183-4\_25, 2012.
- Ewald, F., Bugliaro, L., Mannstein, H., and Mayer, B.: An improved cirrus detection algorithm MeCiDA2 for SEVIRI and its evaluation with MODIS, Atmos. Meas. Tech., 6, 309-322, doi: 10.5194/amt-6-309-2013, 2013.
- Graf, K., Mannstein, H., Mayer, B., and Schumann, U.: 2009: Some evidence of aviation fingerprint in diurnal cycle of cirrus over the North Atlantic. *Proc. 2nd Intern. Conf. Transport, Atmosphere and Climate (TAC-2). 22-25 June 2009, DLR-FB 2010-10, ISSN 1434-8454*, Aachen, 180-185.
- Hamann, U., Walther, A., Baum, B., Bennartz, R., Bugliaro, L., Derrien, M., Francis, P., Heidinger, A., Joro, S., Kniffka, A., Gléau, H. L., Lockhoff, M., Lutz, H.-J., Meirink, J. F., Minnis, P., Palikonda, R., Roebeling, R., Thoss, A., Platnick, S., Watts, P., and Wind, G.: Remote sensing of cloud top pressure/height from SEVIRI: analysis of ten current retrieval algorithms, Atmos. Meas. Tech., 7, 2839-2867, doi: 10.5194/amt-7-2839-2014, 2014.
- Iwabuchi, H., Yang, P., Liou, K. N., and Minnis, P.: Physical and optical properties of persistent contrails: Climatology and interpretation, J. Geophys. Res., 117, D06215, doi: 10.1029/2011JD017020, 2012.
- Kox, S., Bugliaro, L., and Ostler, A.: Retrieval of cirrus cloud optical thickness and top altitude from geostationary remote sensing, Atmos. Meas. Tech., 7, 3233-3246, doi: 10.5194/amt-7-3233-2014, 2014.
- Mannstein, H.: 2005: The ESA-EUROCONTROL 'contrails' project: results. *ATM & Environment Consultation Meeting*, Brussels, Belgium.
- Mannstein, H., and Schumann, U.: Aircraft induced contrail cirrus over Europe, Meteorol. Z., 14, 549 554, 10.1127/0941-2948/2005/0058, 2005.
- Mannstein, H., Meyer, R., and Wendling, P.: Operational detection of contrails from NOAA-AVHRR data, Int. J. Remote Sensing, 20, 1641-1660, doi: 10.1080/014311699212650, 1999.
- Mannstein, H., Brömser, A., and Bugliaro, L.: Ground-based observations for the validation of contrails and cirrus detection in satellite imagery, Atmos. Meas. Tech., 3, 655-669, doi: 10.5194/amt-3-655-2010, 2010.
- Mannstein, H., Vázquez-Navarro, M., Graf, K., Duda, D. P., and Schumann, U.: Contrail detection in satellite images, in: Atmospheric Physics Background Methods Trends, edited by: Schumann, U., Springer, Berlin, Heidelnberg, doi: 10.1007/978-3-642-30183-4\_26, 2012.

- Meyer, R., Mannstein, H., Meerkötter, R., Schumann, U., and Wendling, P.: Regional radiative forcing by line-shaped contrails derived from satellite data, J. Geophys. Res., 107, ACL 17-11 ACL 17-15, 10.1029/2001jd000426, 2002.
- Meyer, R., Buell, R., Leiter, C., Mannstein, H., Pechtl, S., Oki, T., and Wendling, P.: Contrail observations over Southern and Eastern Asia in NOAA/AVHRR data and comparisons to contrail simulations in a GCM, Int. J. Rem. Sens., 28, 2049-2069, doi: 10.1080/01431160600641707, 2007.
- Minnis, P., Palikonda, R., Walter, B. J., Ayers, J. K., and Mannstein, H.: Contrail properties over the eastern North Pacific from AVHRR data, Meteorol. Z., 14, 515-523, doi: 10.1127/0941-2948/2005/0056, 2005.
- Palikonda, R., Minnis, P., Duda, D. P., and Mannstein, H.: Contrail coverage derived from 2001 AVHRR data over the continental United States of America and surrounding areas, Meteorol. Z., 14, 525-536, doi: 10.1127/0941-2948/2005/0051, 2005.
- Schmidt, E.: Die Entstehung von Eisnebel aus den Auspuffgasen von Flugmotoren, in: Schriften der Deutschen Akademie der Luftfahrtforschung, Verlag R. Oldenbourg, München, Heft 44, <a href="http://elib.dlr.de/107948/">http://elib.dlr.de/107948/</a>, edited by, 1-15, 1941.
- Schumann, U.: On conditions for contrail formation from aircraft exhausts, Meteorol. Z., 5, 4-23, doi: 10.1127/metz/5/1996/4, https://elib.dlr.de/32128/, 1996.
- Schumann, U., Mayer, B., Graf, K., Mannstein, H., and Meerkötter, R.: 2009: A parametric radiative forcing model for cirrus and contrail cirrus. *ESA Atmospheric Science Conference, ESA SP-676*, Barcelona, Spain, 7-11 September 2009, 1-6.
- Schumann, U., Hempel, R., Flentje, H., Garhammer, M., Graf, K., Kox, S., Lösslein, H., and Mayer, B.: Contrail study with ground-based cameras, Atmos. Meas. Tech., 6, 3597-3612, doi: 10.5194/amt-6-3597-2013, 2013.
- Strandgren, J., Fricker, J., and Bugliaro, L.: Characterisation of the artificial neural network CiPS for cirrus cloud remote sensing with MSG/SEVIRI, Atmos. Meas. Tech., 10, 4317–4339, doi: 10.5194/amt-10-4317-2017, 2017a.
- Strandgren, J., Bugliaro, L., Sehnke, F., and Schröder, L.: Cirrus cloud retrieval with MSG/SEVIRI using artificial neural networks, Atmos. Meas. Tech., 10, 3547–3573, doi: 10.5194/amt-10-3547-2017, 2017b.
- Vázquez-Navarro, M., Mannstein, H., and Mayer, B.: An automatic contrail tracking algorithm, Atmos. Meas. Tech., 3, 1089–1101, doi: 10.5194/amt-3-1089-2010, 2010.
- Vázquez-Navarro, M., Mayer, B., and Mannstein, H.: A fast method for the retrieval of integrated longwave and shortwave top-of-atmosphere upwelling irradiances from MSG/SEVIRI (RRUMS), Atmos. Meas. Tech., 6, 2627-2640, doi: 10.5194/amt-6-2627-2013, 2013.
- Vázquez-Navarro, M., Mannstein, H., and Kox, S.: Contrail life cycle and properties from 1 year of MSG/SEVIRI rapid-scan images, Atmos. Chem. Phys., 15, 8739-8749, doi: 10.5194/acp-15-8739-2015, 2015.

Zinner, T., Mannstein, H., and Tafferner, A.: Cb-TRAM: Tracking and monitoring severe convection from onset over rapid development to mature phase using multi-channel Meteosat-8 SEVIRI data, Meteorology and Atmospheric Physics, 101, 191-210, doi: 10.1007/s00703-008-0290-y, 2008.

---

## Author Response (AR1)

Dear Editor and Reviewers:

Thank you for the effort in reviewing our manuscript "GVCCS: A Dataset for Contrail and Tracking on Visible Whole Sky Camera Sequences". We appreciate the valuable and constructive feedback provided by both reviewers and the editor, which has significantly helped us improve the clarity, completeness, and context of our work.

We have addressed all comments point-by-point below and revised the manuscript accordingly, with major revisions focusing on clarifying technical terminology, improving the discussion of the dataset's representability, and thoroughly integrating relevant prior work, particularly from the team around Mannstein.

Yours sincerely,
Jarry, G. et al.

**1 Response to reviewer 1**

**General Comment:** This is a well-written paper that provides an extensive introduction to and description of a large, novel observational dataset of contrails. I have mostly minor comments, related to small details in the writing and presentation of the results. These can be found under "line-by-line comments".

> **Comment 1.1**
>
> How were the to-be-labeled video sequences selected? Given a particular sequence start, how was its length determined? How do these choices affect the representability of the dataset, and the generalization performance of algorithms trained with it?

**Response:**

We thank the reviewer for this important question regarding dataset representability and its implications for model generalization. To efficiently construct a dataset rich in contrail instances while maintaining computational feasibility, we employed a two-stage selection process from the complete archive:

1. A lightweight binary classifier was trained to distinguish contrail-present from contrail-absent images. This automated pre-filtering efficiently identified candidate periods by excluding extended intervals of clear sky or heavy low-altitude cloud cover.

2. Final video sequences were manually selected from the filtered periods, prioritizing scenes with visible, persistent contrails suitable for detailed human annotation. This approach maximizes the utility of expert annotation effort and provides sufficient training signal for deep learning models.

We explicitly acknowledge that this strategy introduces a selection bias toward contrail-rich scenarios. The resulting dataset spans the full calendar year, ensuring coverage of diverse seasonal and atmospheric conditions, but over-represents contrail-positive cases relative to the unfiltered archive.

Video length was determined by the temporal span required to capture meaningful contrail evolution within the camera's field of view, from initial formation or entry into the scene through spreading, fragmentation, or dissipation. The duration balances annotation cost with the capture of complete contrail lifecycles for temporal tracking. We have added explicit discussion of the selection methodology and its implications in Sections 4.1 and 6.1 of the revised manuscript.

> **Comment 1.2**
>
> Do you expect that the video-based models would perform better as the length of the clips used for training them is increased? If I interpret the paper correctly, the maximum length of video clips that are used is 5 frames (so 2.5 minutes). This seems to be quite short, especially when placed in comparison to the length of temporal contexts used when analyzing geostationary satellite imagery (e.g. Ng et al. 2023).

**Response:**

We thank the reviewer for this insightful question. The reviewer is correct that our video-based models use relatively short clips (3-5 frames, corresponding to 1.5-2.5 minutes) due to GPU memory constraints during training and inference.

We initially hypothesized that longer temporal contexts would improve performance by providing richer motion cues and better instance consistency. However, our results (Table 2) show that the video-based models do not consistently outperform image-based models on per-frame metrics. The Swin-L image-based variant achieves the highest overall instance segmentation performance (AP@[0.25:0.75 | all | 100] = 0.37 for single-polygon), slightly exceeding its video counterpart (0.34). This suggests that, at least within the clip lengths tested, temporal information does not uniformly translate into improved per-frame accuracy.

Several factors may contribute to this outcome. First, the image and video models were initialized from different pretrained checkpoints: COCO (images, general objects) versus YouTubeVIS (videos, motion-focused). These pretraining differences may influence downstream performance independent of temporal modeling. Second, the video model must jointly optimize for segmentation accuracy and temporal consistency, which may introduce trade-offs not present in single-frame prediction. Third, as noted in Section 6.2, our evaluation metrics are computed per-frame and do not reward temporal coherence, an area where video models provide clear qualitative benefits not captured by the standard average precision (AP) metric.

While satellite-based studies (e.g., Ng et al. 2023) use longer sequences (frame count), the spatial and temporal scales differ substantially. Geostationary satellites observe contrails at coarser resolution over much larger geographic areas, where contrails evolve more slowly in image space. Our ground-based camera captures high-resolution imagery with 30-second sampling, providing finer spatial detail but faster apparent motion due to proximity and perspective effects. Given these

differences in resolution, field of view, and contrail dynamics, we believe the temporal contexts are functionally comparable despite the difference in frame count.

We are currently investigating architectures capable of processing longer sequences to determine whether extended temporal context yields measurable improvements. Preliminary experiments are underway, and we will report findings in future work. We emphasize that the primary contribution of this work is the dataset itself, not achieving state-of-the-art model performance. The baseline models presented serve to demonstrate the dataset's utility and to establish reproducible benchmarks for the community.
* * *
**Comment 1.3**

It is found that the image-based models show better instance-segmentation performance than the video-based models, but that the latter also provides "tracking" output. What is the quality of these "tracking" outputs and is it sufficiently better than what could be obtained from the "better instance segmentation" output in combination with a simple tracking algorithm (e.g. based on overlap between image frames).
* * *
**Response:**

We thank the reviewer for this important observation regarding the trade-offs between the image-based and video-based approaches. The reviewer correctly identifies that image-based models could theoretically be combined with post-hoc tracking algorithms (e.g., IoU-based matching between frames) to achieve temporal consistency. Indeed, we implement precisely such an approach in Appendix A, where we apply a Hungarian algorithm-based tracking method to link instances across frames using overlap similarity. This demonstrates that tracking can be achieved through post-processing of image-based predictions. However, the key distinction lies in *how* tracking is achieved. The video-based model learns temporal associations end-to-end during training, jointly optimizing segmentation accuracy and temporal consistency within a unified objective. In contrast, post-hoc tracking methods operate independently of the segmentation model and must infer correspondences from spatial overlap alone, without access to learned motion cues, appearance features, or temporal context that inform the video model's predictions.

While we acknowledge that our current evaluation uses per-frame metrics that do not reward temporal consistency, the video-based approach offers several practical benefits: (1) reduced instance ID flickering and fragmentation across frames, (2) more robust handling of occlusions and brief disappearances, and (3) implicit learning of contrail motion patterns from data. These qualities are particularly valuable for downstream applications such as lifecycle analysis and contrail-flight attribution, where maintaining stable instance identities over extended periods is critical.

We emphasize that the primary contribution of this paper is the GVCCS dataset, not achieving optimal model performance. Both the image-based and video-based models serve as baseline demonstrations to showcase the dataset's utility for different use cases. The image-based approach may be preferred when computational resources are limited or when per-frame accuracy is paramount, while the video-based approach is better suited to applications requiring continuous tracking. We provide both options and leave the choice to practitioners based on their specific needs.

Future work should indeed conduct systematic comparisons of tracking quality using video-specific metrics (e.g., tracking accuracy, ID switches, fragmentation). However, such an analysis is beyond the scope of this dataset paper. We have added clarifying text in Section 6.2 to acknowledge this limitation and to note that the temporal consistency benefits of video models are not fully captured by our frame-level evaluation metrics.
* * *
**Comment 1.4**

Could more examples of how the image pre-processing technique helps identify contrails, be given? Currently, only 1 such result is included, but more examples that show the value of the re-projection and contrast enhancement would be helpful in my opinion.
* * *
**Response:**

We thank the reviewer for this suggestion to better illustrate the value of our preprocessing pipeline.

The geometric projection and enhancement steps serve two primary purposes: (1) facilitating accurate human annotation by presenting contrails in a more natural, undistorted view, and (2) simplifying the learning task for computer vision models by removing lens-induced distortions and enhancing visual contrast.

The geometric projection transforms the fisheye image into a planar grid at constant altitude (10 km), converting curved contrails that appear highly distorted near the image periphery into approximately linear structures. This linearization is crucial because contrails are inherently straight or gently curved in physical space. The apparent curvature in raw fisheye imagery is purely an artifact of the lens geometry. By removing this artificial distortion, annotators can more easily delineate contrail boundaries, and models can leverage the natural linear structure of contrails without learning camera-specific deformations.

The three-step enhancement pipeline (brightness scaling, CLAHE, and color rebalancing) significantly improves contrail visibility, particularly in challenging conditions such as high solar glare, thin cirrus backgrounds, or low-contrast atmospheric scenes. Without enhancement, many faint or partially formed contrails would be difficult for annotators to identify consistently, potentially leading to incomplete or inconsistent labels.

To better demonstrate these benefits, we have expanded Figure 2 in Section 4.1 to include the three steps. Now, this example illustrates how the preprocessing pipeline enables high-quality annotation and robust model training.

We emphasize that all models in this study are trained and evaluated exclusively on preprocessed images. The preprocessing is therefore an integral part of the dataset, not an optional step, and its benefits are implicitly reflected in the model performance reported throughout the paper.

**Comment**

Line-by-line comments:

- P1L6: "don't" -> "do not", "aspect" -> "aspects"

- P1L21: "trap outgoing" -> "reduce outgoing"

- P1L22: "as due to aviation CO2 emissions"

- P2L30: "context-dependent and extremely difficult to model reliably" -> "highly variable and challenging to model."

- P2L33: APCEMM stands for "Aircraft Plume Chemistry, Emissions, and Microphysics Model"

- P2L53: I think it would be nice for the authors to explain in more detail why the attribution of observed contrails to flights is difficult when using geostationary imagery, and how the introduced dataset does not suffer from this problem.

- P2L56: I think I understand what this paragraph is trying to say, but I feel like it could be made clearer. Perhaps state this as, "existing datasets of contrails annotated in observational data such as the OpenContrails dataset, do not track individual contrails over time or provide information on the flights that formed them"?

- P3L70: is there no reference for this Mask2Former approach?

- P4L106: Although named after Schmidt and Appleman, the criterion in its current form was originally presented by (Schumann, 1996) so I would suggest also citing that paper here.

- P4L109: "trap" the more physically correct term is "reduce".

- P4L111: "The precise relative impact" what exactly is meant here with "relative impact"?

- P4L117: "Atmospheric imagery" what is meant with this term?

- P5L131: "coverage" -> "spatial coverage"

- P8L198: the labeling of these images was done at the "semantic segmentation" level, not at the instance segmentation level.

- P8L205: "Thanks to the 10-minute temporal..." This sentence does not make much sense to me. Perhaps it could be left out, as the rest of the paragraph is clear.

- P8L209: "A 2025 update" Technically, the instance-level labels were always there, but not released as part of the 2023 Kaggle competition that utilized the OpenContrails dataset. So I would suggest to leave out this sentence.

- P8L210: The authors could also cite (Pertino et al., 2024) here.

- P8L219: "of dataset" –> "of a dataset". The authors should note that earlier studies have collocated contrails between different remote sensing instruments. Examples are (Iwabuchi et al., 2012; Mannstein et al., 2010; Vazquez-Navarro et al., 2010).

- P8L220: "Altitude" appears twice in this sentence.

- P9L244: "Non-data-driven image-analysis techniques" -> "Traditional image analysis techniques"

- P9L254: Citation "Jarry et al." missing a year. Happens elsewhere as well.

- P10L266: "Leveraging as well on Hough-based line detection" This sentence doesn't make sense to me.

- P10L282: provide exact coordinates of camera location, if possible.

- P12L315: "or reviewing" -> "of reviewing"

- P14L371: "U-net" written differently than elsewhere in the paper

- P14L387: "k-means" maybe write "k" in italic?

**Comment**

Line-by-line comments (continuation):

- P23 Figure 5: Include whether time is "UTC" or not. Same goes for Figure 7. Additionally, "Raw image" is used for an image that has already undergone quite some processing, so perhaps a different terminology could be used here?

- P24 Figure 6: I think it would be helpful to combine figures 5 and 6 to make it easier for the reader to perform the visual comparisons.

- P24L594: "erroneously merges contrails 5 and 6 into a single prediction" I don't see this in the results at all? Is this potentially a typo? Should it be contrails 3 and 6?

- P24L595: "merges contrails 5 and 6" Again, I don't see this.

- P26L638: "we are deployed" -> "we are deploying" ?

- P26L656: "Integrating these tasks . . ." This sentence doesn't make sense to me.

**Response:**

We appreciate the detailed line-by-line comments, which significantly help improve the clarity and precision of the manuscript. We have implemented all requested changes as listed below.

- P1L6: Changed "don't" to "do not" and "aspect" to "aspects".

- P1L21: Changed "trap outgoing" to "reduce outgoing".

- P1L22: Corrected the phrase to "as that from CO2 emissions".

- P2L30: Changed the phrasing to "highly variable and challenging to model."

- P2L33: Defined APCEMM as "Aircraft Plume Chemistry, Emissions, and Microphysics Model".

- P2L53: We have expanded the discussion in Section 2.1 to better explain the challenges of contrail-flight attribution in satellite imagery versus ground-based observations.

- P2L56: Revised the sentence for clarity, aligning with the suggested "existing datasets of contrails annotated in observational data such as the OpenContrails dataset, do not track individual contrails over time or provide information on the flights that formed them."

- P3L70: Added the relevant citation for the Mask2Former approach.

- P4L106: Added citation for (Schumann, 1996) to correctly reflect the history of the Schmidt-Appleman criterion.

- P4L109: Changed "trap" to the more physically correct term "reduce".

- P4L111: By "relative impact" we meant the magnitude of contrail climate forcing relative to (i.e., compared with) aviation's $CO_2$ emissions. We have revised the text to remove this ambiguous phrase and state more clearly that the comparison depends on the climate metric used (e.g., effective radiative forcing vs. global warming potential vs. temperature response).

- P4L117: Clarified "Atmospheric imagery" to mean imagery from "satellite and ground-based remote sensing instruments".

- P5L131: Changed "coverage" to "spatial coverage".

- P8L198: Clarified that the dataset labeling was done at the "semantic segmentation" level.

- P8L205: Removed the confusing sentence "Thanks to the 10-minute temporal. . .".

- P8L209: Removed the sentence about the "A 2025 update" to avoid confusion.

- P8L210: Added citation for (Pertino et al., 2024).

- P8L219: Corrected "of dataset" to "of a dataset" and added a note and citations for earlier collocated contrail studies: (Iwabuchi et al., 2012; Mannstein et al., 2010; Vazquez-Navarro et al., 2010).

- P8L220: Corrected the sentence to remove the duplicate word "Altitude".

- P9L244: Changed "Non-data-driven image-analysis techniques" to "Traditional image analysis techniques".

- P9L254: Added the year to the "Jarry et al." citation and checked other instances.

- P10L266: Rephrased the sentence about Hough-based line detection for better readability.

- P10L282: Provided the exact coordinates of the camera location in the text.

- P12L315: Changed "or reviewing" to "of reviewing".

- P14L371: Uniformed the writing of U-Net.

- P14L387: Changed "k-means" to $k$-means (using italic $k$).

- P23 Figure 5: Included a note in the caption that the time is UTC and changed "Raw image" to "Original projected and enhanced image" for more accurate terminology in the figure and caption text.

- P24 Figure 6: While acknowledging the suggestion, we will maintain separate figures but ensure close cross-referencing to facilitate visual comparison.

- P24L594, P24L595: Confirmed the suggested typo; changed "contrails 5 and 6" to "contrails 3 and 6" in both instances.

- P26L638: Corrected "we are deployed" to "we are deploying".

- P26L656: Rephrased the sentence "Integrating these tasks . . . " for clarity and flow.

**2 Response to reviewer 2**

**General Comments:**

The paper deals with an important task: The setup of an observational database of contrail properties suitable for checking the validity of contrail model results. The paper describes the dataset "GVCCS: Ground Visible Camera Contrail Sequences repository" which one (including me) could download from https://doi.org/10.5281/zenodo.16419651. The dataset contains video pictures with very many whole-sky camera figures showing all kinds of clouds including contrails and a set of parquet files which I cannot read because I have no Python knowledge (which tool could one use to open these files and to look at these data???). The paper gives a technical description of how the files are produced and handled. Also, the paper describes 1. the goals, 2. some of the background knowledge, 3. the state of the art, 4. the datasets, 5. the segmentation models, 6. the results, and 7. the conclusions.- This structure obviously contains some overlaps (e.g., background and state of the art). The method provides a "segmentation" which means a list of segment elements with their basic properties, such as representing a contrail. The method identifies boxes containing contrails and "Instance segmentation" which I learned here that this means the pixels containing contrail cloud elements and allows for "panoptic segmentation" which assigns each pixel to a unique contrail. (I must say, I am not quite sure that I have understood this correctly; these terms are new for me). For me the paper was hard to read and I did by far not understand all details. I can say that the paper is worth to have and it should be published. However, I recommend that the authors go through the paper together with somebody who has no or only little experiences in all the tools and terms used, to make sure that all terms used are well defined and to improve the readability.
* * *
**Comment 2.1**

I miss a discussion of several older papers on this and related topics. For example, the paper misses work of the team around Mannstein (-2013). You cite one of his papers (Mannstein et al., 1999) but there are several others: (Meyer et al., 2002; Mannstein, 2005; Mannstein and Schumann, 2005; Minnis et al., 2005; Palikonda et al., 2005; Meyer et al., 2007; Zinner et al., 2008; Graf et al., 2009; Schumann et al., 2009; Mannstein et al., 2010; Mannstein et al., 2012; Ewald et al., 2013; Vázquez-Navarro et al., 2013; Vázquez-Navarro et al., 2015). I recommend to check also the following papers for related work: (Bugliaro et al., 2012; Iwabuchi et al., 2012; Hamann et al., 2014; Kox et al., 2014; Strandgren et al., 2017b; Strandgren et al., 2017a). Some of these references should be included in the Introduction, for example at line 40/41.
* * *
**Response:**

We sincerely thank the reviewer for this comprehensive and vital list of references, particularly the foundational work from the team around Mannstein and colleagues. We fully agree that these papers are essential for providing complete historical and technical context for contrail detection and characterization, and we acknowledge that our original manuscript did not sufficiently recognize this extensive body of pioneering research. We have thoroughly reviewed all suggested references and integrated them throughout the revised manuscript. Specifically, we have:

- Expanded the Introduction (Section 1) to properly acknowledge the foundational satellite-based contrail detection methods, as well as ground-based validation studies.

- Strengthened Section 2.1 (Background on Contrails) by incorporating key studies on contrail radiative forcing, contrail properties and climatology, and the relationship between contrail formation and atmospheric conditions.

- Enhanced Section 3 (State of the Art - Models) to include the historical progression of contrail detection algorithms, from early Hough-transform and threshold-based approaches through advanced cirrus and contrail detection methods, to modern deep learning techniques. This provides essential context showing how the field has evolved from classical computer vision to data-driven approaches.

These additions substantially strengthen the manuscript by properly situating our work within the rich history of contrail remote sensing and demonstrating how ground-based, high-resolution video datasets complement the extensive satellite-based research pioneered by these groups. We thank the reviewer for ensuring we provide appropriate recognition of this foundational body of work.
* * *
**Comment 2.2**

Line 106: Appleman did not explain the process completely. For example, the impact of engine efficiency was not understood by him. Schmidt (1941) was the first who understood that contrails form when the ambient temperature is low enough so that the humidity inside the plume gets high enough to let water condense. Schumann (1996) covers the concept more completely, including engine efficiency and practical analysis methods, and introduced the term "Schmidt-Appleman Criterium".

**Response:**

We thank the reviewer for this important historical clarification. The reviewer is correct that Appleman's work, while foundational, did not fully capture the underlying physics, particularly the role of engine efficiency. Schmidt (1941) was indeed the first to properly explain that contrails form when ambient temperature is sufficiently low to cause high humidity in the aircraft plume, leading to water condensation. Schumann (1996) later provided a comprehensive treatment that incorporated engine efficiency, developed practical analysis methods, and formalized the term "Schmidt-Appleman Criterion" that is widely used today. We have revised the text in Section 2.1 to accurately reflect this historical progression and have updated the citations to properly credit Schmidt (1941) and Schumann (1996) alongside Appleman (1953).
* * *
**Comment 2.3**

Line 119: See also Mannstein et al. (2010).
* * *
**Response:**

We have added the citation to Mannstein et al. (2010) at Line 119 as suggested to provide further relevant context.
* * *
**Comment 2.4**

Line 143/328: Where can I read more about COCO?
* * *
**Response:**

COCO stands for Common Objects in Context. It is a widely used large-scale benchmark dataset in computer vision for tasks including object detection, instance segmentation, and captioning. It is the gold standard for comparing the performance of segmentation models. We have added a citation to the original paper (Lin et al., 2014) wherever COCO is mentioned to allow readers to learn more about its structure and metrics.
* * *
**Comment 2.5**

Line 145: If you say "traditionally", then there should be lots of examples in the literature. But I miss citations.
* * *
**Response:**

We thank the reviewer for pointing out this omission. Traditional object detection methods that rely on axis-aligned bounding boxes include widely used approaches such as Faster R-CNN and YOLO, both of which have become standard baselines in computer vision. We have added these citations to the revised manuscript at line 145 to support the discussion of traditional bounding box-based detection methods.
* * *
**Comment 2.6**

Line 151: Instance segmentation was done, e.g. by Mannstein et al. (1999). How would you classify the methods used by Schumann et al. (2013)?
* * *
**Response:**

We thank the reviewer for this important clarification. The reviewer is correct that Mannstein et al. (1999) performed what we would now classify as instance segmentation. Their method identified contrail pixels and grouped spatially connected regions into distinct contrail objects, effectively separating individual contrails.

The work by Schumann et al. (2013) employs ground-based cameras with automated detection algorithms that identify and track individual contrails, characterize their properties (width, optical depth, coverage), and validate these measurements against radiative transfer modeling. Their approach combines automated detection with physical characterization and thus also performs instance-level contrail identification, as individual contrails are tracked and their properties measured separately.

We acknowledge that our manuscript's framing in Section 2.2 may have inadvertently suggested that instance segmentation is a recent innovation, when in fact the atmospheric science community has been performing instance-level contrail detection and characterization for decades, albeit using different methodological frameworks (feature-based detection, connectivity analysis, trajectory matching) than modern deep learning approaches (end-to-end neural networks with learned representations).

We have revised Section 2.2 to clarify that instance segmentation as a *task* (separating individual objects) has long been addressed in contrail research, while our work focuses on modern deep learning architectures (Mask2Former) that perform this task through learned feature representations.
* * *
**Comment 2.7**

Line 219: Not only Meijer but others did similar work: E.g., Vázquez-Navarro et al. (2010) first identified contrail segments in high-resolution Low- Earth-Observation data (MODIS) and then followed the contrails in time in a sequence of Meteosat data (with high time resolution). I assume that has been done similarly also by others.

**Response:**

We appreciate the suggestion and have revised the text at Line 219 to acknowledge that earlier studies have also collocated contrail observations across different remote sensing platforms. We have added the reference to Vázquez-Navarro et al. (2010), highlighting the importance of combining high-resolution spatial and high-resolution temporal data sources.
* * *
**Comment 2.8**

Line 244: What is the meaning of "non-data driven"?

**Response:**

The term "non-data driven" refers to traditional methods that rely on pre-defined mathematical rules, fixed thresholds, or hand-crafted image processing algorithms (e.g., edge detection, morphological operations) to process data, rather than learning patterns and features from a large training dataset (which is "data-driven" machine learning). As suggested by Reviewer 1, we have replaced this term with the clearer phrase "Traditional image analysis techniques" and provided this definition.
* * *
**Comment 2.9**

Line 294: What is CLAHE - is there a reference?

**Response:**

CLAHE stands for Contrast Limited Adaptive Histogram Equalization. It is a standard image processing technique used to improve image contrast by operating on small regions (tiles) of the image, rather than the entire image, thereby preventing over-amplification of noise. We have defined the acronym in the text.
* * *
**Comment 2.10**

Line 333: What is a U-Net?

**Response:**

A U-Net is a type of Convolutional Neural Network (CNN) specifically designed for semantic segmentation. Its name comes from its distinct U-shaped architecture, which includes a contracting path (encoder) that captures context and an expansive path (decoder) that enables precise localization. A key feature is the use of skip connections that pass information from the encoder to the decoder to preserve fine-grained details lost during downsampling. We have added this explanation and the original citation (Ronneberger et al., 2015).
* * *
**Comment 2.11**

Line 350: What is a loss-model?

**Response:**

We apologize for the unclear terminology. In machine learning, a *loss function* (also called objective function or cost function) is a mathematical function that quantifies the difference between a model's predictions and the ground truth annotations. During training, the model iteratively adjusts its parameters to minimize this loss function, thereby improving prediction accuracy. In the context of segmentation, the loss function measures how well predicted masks match the annotated masks, penalizing both false positives (predicted contrail pixels that are not contrails) and false negatives (missed contrail pixels).

We have clarified this terminology in the revised manuscript, replacing "loss-model" with "loss function" and adding a brief explanation where it first appears to make the text accessible to readers unfamiliar with machine learning terminology.

**Comment 2.12**

Line 355: What is a Hungarian algorithm?

**Response:**

The Hungarian Algorithm is a combinatorial optimization method used to solve the assignment problem. In the context of our detection and tracking model, it is used to find the optimal one-to-one matching between the set of predicted object instances (contrails) and the set of ground-truth instances from the annotations. This matching is crucial for calculating the loss function and training the model efficiently.

**Comment 2.13**

Line 376: What are skip connections?

**Response:**

Skip connections (also called shortcut connections or residual connections) are direct pathways in neural network architectures that bypass one or more layers, allowing information to flow from earlier layers directly to deeper layers. In the U-Net architecture specifically, skip connections link corresponding layers in the encoder (downsampling path) and decoder (upsampling path), allowing high-resolution spatial details captured early in the network to be combined with the semantic information extracted at deeper layers. This is crucial for segmentation tasks because the encoder's downsampling operations progressively discard fine-grained spatial information to build abstract representations; skip connections enable the decoder to recover this lost detail when reconstructing the pixel-level output. Without skip connections, the segmentation masks would lack the precise boundary localization needed for accurate contrail delineation.

We have added a clearer explanation of skip connections in Section 5.2 where the U-Net architecture is introduced.

**Comment 2.14**

Line 389: What is the HDBSCAN algorithm?

**Response:**

HDBSCAN (Hierarchical Density-Based Spatial Clustering of Applications with Noise) is a clustering algorithm that groups data points based on local density, identifying clusters of arbitrary shape without requiring a predetermined number of clusters. Unlike k-means, which assumes spherical clusters and requires specifying the number of clusters in advance, HDBSCAN builds a hierarchical tree of potential clusters and extracts stable clusters across multiple density scales. It also identifies noise points (outliers) that do not belong to any cluster. In our instance segmentation pipeline, HDBSCAN is applied to the pixel embeddings learned by the U-Net: pixels with similar embeddings (high local density in embedding space) are grouped into the same contrail instance, while isolated pixels are treated as outliers. This approach is well-suited to contrails, which may have irregular, fragmented shapes that traditional clustering methods struggle to handle.

We have added a citation and brief explanation of HDBSCAN in Section 5.2 of the revised manuscript.

**Comment 2.15**

Line 410: I have some general knowledge on Principle Component Analysis, but I do not see how this is applied here.

**Response:**

We thank the reviewer for seeking clarification on this point. In the U-Net discriminative embedding approach, each pixel is represented by a high-dimensional feature vector (embedding), typically with 16, 32, or more dimensions. These embeddings exist in a high-dimensional space where pixels belonging to the same contrail instance are clustered together. To visualize this high-dimensional embedding space in Figure 4, we apply Principal Component Analysis (PCA) to reduce the dimensionality from the original embedding dimension (e.g., 32D) to just 2 dimensions for plotting. PCA identifies the two orthogonal directions in the embedding space that capture the most variance, effectively projecting the complex high-dimensional clusters onto a 2D plane that can be displayed in a figure. Each point in the visualization represents a single pixel,

colored according to its ground-truth instance label, allowing readers to visually assess whether pixels from the same contrail instance are indeed clustered together in the learned embedding space.

We have clarified this explanation in the caption and text accompanying Figure 4 to make the role of PCA more explicit.
* * *
**Comment 2.16**

Fig. 4, caption: When is a label "true"? What is "discriminative embedding"???
* * *
**Response:**

We apologize for the confusing terminology. By "true label" we mean the ground-truth annotation—the human-labeled instance segmentation that serves as the reference for training and evaluation. "Discriminative embedding" refers to the learned feature representation where each pixel is mapped to a point in a high-dimensional space such that pixels belonging to the same contrail instance are close together (forming a cluster), while pixels from different instances are far apart. The term "discriminative" indicates that the embedding is trained to discriminate between (i.e., separate) different instances.

We have completely rewritten the caption for Figure 4 to eliminate jargon and clearly explain what each panel shows, using accessible language suitable for readers unfamiliar with machine learning terminology.
* * *
**Comment 2.17**

Chapter 6: Chapter 6 is indigestible, at least for me. It needs careful revision.
* * *
**Response:**

We thank the reviewer for this feedback. Following the detailed comments from both reviewers, we have substantially revised Section 6 to improve accessibility, including: (i) clear definitions of machine learning terminology unfamiliar to atmospheric scientists (loss functions, Hungarian algorithm, HDBSCAN, discriminative embeddings, etc.); (ii) enhanced explanations of evaluation metrics and their limitations; (iii) improved figure captions with detailed, self-contained explanations; and (iv) additional context on model trade-offs and dataset construction implications.

We believe these revisions have significantly improved readability while maintaining the technical rigor necessary for reproducibility. Section 6 necessarily contains substantial technical detail to enable the community to build upon this work, but we have structured the presentation to progressively build understanding from simpler concepts (semantic segmentation) to more complex ones (instance segmentation, tracking), concluding with qualitative examples that contextualize the quantitative results.
* * *
**Comment 2.18**

Page 23: Where is Fig. 6 mentioned?
* * *
**Response:**

We thank the reviewer for this observation. In the original submission, Figure 6 was indeed not properly referenced in the text—an oversight on our part. In the revised manuscript, we have restructured the figures section and this particular figure has been removed as it was redundant with information already presented in the tables and other figures. All remaining figures in the revised manuscript are now explicitly referenced in the main text.
* * *
**Comment 2.19**

Line 619: What is a Dice score and what is mIoU, and what is AP@[0.25:0.75 etc. meaning?
* * *
**Response:**

We apologize for assuming familiarity with these computer vision metrics. We have added detailed explanations in Section 6.2 (Evaluation) to define these terms:

- **Dice coefficient (Dice score):** Measures overlap between predicted and ground-truth masks as $2\times(\text{intersection})/(\text{prediction size}+ \text{ground truth size})$. Values range from 0 (no overlap) to 1 (perfect match). It emphasizes correct overlap and is particularly sensitive to small or thin objects like contrails.

- **mIoU (mean Intersection over Union):** Measures overlap as intersection/union of predicted and ground-truth masks. It penalizes both false positives and false negatives equally. Values range from 0 to 1.

- **AP@[0.25:0.75 | all | 100]:** Average Precision computed across IoU thresholds from 0.25 to 0.75, considering all object sizes, with a maximum of 100 detections per image. This notation indicates: [IoU threshold range | size category | max detections]. We use a relaxed IoU range (0.25-0.75 instead of the standard COCO 0.50-0.95) because contrails' thin, elongated geometry makes very high IoU thresholds overly strict.

These explanations are now integrated into Section 6.2 where the metrics are first introduced, with sufficient detail to interpret the results without requiring prior computer vision expertise.
* * *
**Comment 2.20**

Line 638: One word seems to be too much in "we are deployed"

**Response:**

We agree. We have corrected the phrase at Line 638 from "we are deployed" to the clearer and grammatically correct phrase "we are deploying".
* * *
**Comment 2.21**

Line 665: I have no idea what an "data-driven ecosystem" could be.

**Response:**

We thank the reviewer for pointing out this unclear jargon. By "data-driven ecosystem" we meant a collaborative research framework where observational datasets, physical models, and machine learning tools are openly shared and integrated to advance contrail research. We have replaced this vague phrase with clearer, more concrete language in the revised manuscript, specifying that we envision a collaborative approach combining observational data with physical modeling across multiple research groups and geographic locations.
* * *
**Comment 2.22**

Very minor detail: Line 80: Unnecessary comma after the 5th word.

**Response:**

We have corrected this minor detail and removed the unnecessary comma at Line 80.
* * *
**Comment 2.23**

At the end you might also discuss the idea of using several cameras distributed in sufficient neighborhood such that each contrail is seen by 2 cameras. This would allow for stereographic height analysis.

**Response:**

We thank the reviewer for this excellent suggestion. Stereographic height determination using multiple ground-based cameras is indeed a promising approach that could complement our single-camera methodology. We have added a discussion of this possibility in the Conclusions section (Section 7), noting that a network of spatially distributed cameras observing the same contrails from different viewing angles would enable triangulation-based altitude estimation without relying on assumed contrail altitudes. This would provide direct observational validation of contrail formation heights, which is currently one of the key uncertainties in contrail modeling. We acknowledge this as an important direction for future work that would significantly enhance the scientific value of ground-based contrail monitoring systems.